

# Similitude of ice-sheet dynamics against scaling of geometry and physical parameters

**J. Feldmann**[1,2] **and A. Levermann**[1,2,3]

[1]Potsdam Institute for Climate Impact Research (PIK), Potsdam, Germany
[2]Institute of Physics, University of Potsdam, Potsdam, Germany
[3]LDEO, Columbia University, New York, USA

*Correspondence to:* A. Levermann (anders.levermann@pik-potsdam.de)

**Abstract.** The concept of similitude is commonly employed in the fields of fluid dynamics and engineering where scaling laws are derived from the governing equation of flow dynamics, e.g., the Navier-Stokes equation. Here we transfer this method to the problem of ice-sheet flow to examine the dynamic similitude of ice sheets against the scaling of their geometry and physical parameters. Carrying out a dimensional analysis of the stress balance for isothermal ice sheets in shallow-shelf approximation we obtain dimensionless numbers that characterize the flow, similar to the Reynolds or Froude numbers in fluid dynamics. Requiring that these numbers remain constant under scaling we obtain conditions that relate the geometric scaling factors, the parameters for the ice softness, surface mass balance and basal friction as well as the ice-sheet intrinsic response time to each other. We demonstrate that these scaling laws are the same for both the (two-dimensional) flow-line case and the three-dimensional case and that they are consistent with flow-line boundary-layer theory. The theoretically predicted ice-sheet scaling behavior agrees with results from numerical simulations that we conduct in flow-line and three-dimensional conceptual setups. In a set of experiments the setup geometry is scaled systematically and the physical parameters are prescribed according to the derived scaling laws. We further investigate analytically the implications of geometric scaling of ice sheets for their response time under constant basal conditions finding that thicker (thinner) ice sheets have a shorter (longer) response time and that the opposite holds for the horizontal ice-sheet extent. With this study we provide a framework which, under several assumptions, allows for a fundamental comparison of the ice-dynamic behavior across different scales. It proofs to be useful in the design of conceptual model setups but might also be applied to real-world

systems, e.g., to examine the response times of glaciers, ice streams or ice sheets to climatic perturbations.

# 1 Introduction

In the fields of fluid dynamics and engineering scaling laws are used to perform experiments with spatially reduced models in water channels or wind tunnels to predict the behavior of the associated full-scale system (Scruton, 1961; Li et al., 2013). Dimensional analysis and the principle of similitude allow to derive such scaling laws analytically (Rayleigh, 1915; Macagno, 1971; Szücs, 1980). For instance, a dimensional analysis of the Navier-Stokes equation (Kundu et al., 2012) yields the Reynold's number (Reynolds, 1883) as one of the dimensionless parameters of the governing equation which characterize the dynamics of fluid flow. Under the assumption of the similitude principle the Reynold's number can provide a scaling law for the fluid's characteristic linear dimension, velocity and viscosity that assures similar flow patterns. The principle of similitude is applied well beyond the field of engineering, e.g. in zoology (land mammals move in dynamically similar fashion at equal Froude number, Alexander and Yayes, 1983) or biology (Stahl, 1962).

Here we apply the concept of similitude to ice-sheet dynamics. Our investigation is based on the shallow-shelf approximation (SSA, Morland, 1987; MacAyeal, 1989; Greve and Blatter, 2009)) of the full-Stokes stress balance. Neglecting the terms of vertical shearing in the stress balance and accounting for the small thickness-to-length ratio of ice sheets, the SSA represents the relevant dynamics of floating ice shelves and grounded ice streams, i.e., regions that are characterized by fast plug-like flow, which has been shown in



numerical applications (Goldberg et al., 2009; Gudmundsson et al., 2012). The SSA can be complemented by the shallow-ice approximation (Hutter, 1983; Huybrechts, 1990; Sato and Greve, 2012) to also include vertical shearing, which is dominant in the more stagnant interior parts of an ice sheet (Bueler and Brown, 2009; Pollard and DeConto, 2012; Thoma et al., 2015), whereas higher-order approximations (Schoof and Hindmarsh, 2010; Larour et al., 2012; Cornford et al., 2015) neglect less stress components in the full-Stokes stress balance (Favier et al., 2012). The MISMIP3d benchmark revealed that numerical models applying the SSA can capture grounding-line dynamics comparable to more elaborate models in conceptual experiments (Pattyn et al., 2013; Feldmann et al., 2014) which will be put to test in the forthcoming MISMIP+ intercomparison project (Asay-Davis et al., 2015).

A dimensional analysis of the ice-dynamic equations is often carried out to compare the magnitudes of the different acting forces and thus to derive physically motivated approximations, as done when deriving the SSA from the full-Stokes stress balance (MacAyeal, 1989; Greve and Blatter, 2009). The non-dimensionalized form of the SSA itself and the involved dimensionless coefficients that result from the introduction of typical scales for, e.g., ice-sheet thickness and velocity, have been used to consider asymptotic limits of SSA ice-sheet flow in previous work (Schoof, 2007a; Dupont and Alley, 2005; Tsai et al., 2015; Haseloff et al., 2015). In the present study we utilize these coefficients to derive ice-sheet scaling laws for the geometry, response time and other physical ice-sheet parameters, a step that to our knowledge, has not been taken before. The scaling behavior of ice sheets, that here is analyzed in a conceptual way, might be of use to better understand the large-scale evolution of the polar ice sheets. Of particular interest is the scaling of the ice-sheet response time (Levermann et al., 2013, 2014) against the background of Antarctic instabilities (Weertman, 1974; Schoof, 2007b; Rignot et al., 2014; Fogwill et al., 2014; Mengel and Levermann, 2014). The time scales of possible rapid ice discharge due to instability in the past (Pollard and DeConto, 2009; Pollard et al., 2015) and future (Favier et al., 2014; Joughin et al., 2014; Feldmann and Levermann, 2015b) are highly uncertain.

The paper is structured as follows: In the next section the governing equations in SSA are non-dimensionalized to derive ice-sheet scaling laws for one and two horizontal dimensions, respectively. We also give an alternative approach to derive the same scaling conditions. Afterwards the analytically predicted ice-sheet scaling behavior is compared with results from numerical modeling. To this extent conceptual experiments are designed in two and three spatial dimensions. Steady states as well as the transient response to perturbation of the simulated ice sheet are analyzed for a systematic variation of the scaling parameters which are prescribed according to the scaling laws. We then examine analytically the implications of the scaling conditions for the response

times of ice sheets considering the geometric scaling factors and basal friction parameter as independent variables. Eventually we discuss the results and conclude.

## 2 Similarity of shallow ice-sheet dynamics

Here we derive scaling laws that determine how the geometry, response time and the involved physical parameters for ice softness, surface mass balance and basal friction have to relate in order to satisfy similitude between different ice sheets. This is visualized conceptually in Fig. 1 for two ice sheets which differ in vertical and horizontal scale. Based on the governing equations in dimensionless form, we obtain dimensionless scale factors which depend on the scales of the geometric and physical parameters of the ice sheet. The requirement that each of these factors has to remain constant under a scaling of the parameters makes sure that the dynamic equations remain exactly the same. The resulting scaling laws thus put constraints on the parameter scaling, ensuring similitude between the different ice-sheet configurations.

### 2.1 Basic equations for similitude analysis

The problem addressed here is the one of an isothermal ice-sheet in SSA (Greve and Blatter, 2009). The two horizontal components of the stress balance in SSA with spatially uniform ice softness $A$ are given by

$$
\begin{aligned}
&A^{-1/n} \left( \frac{\partial}{\partial x} \left[ H \dot{\epsilon}_e^{1/n-1} \left( 2\frac{\partial v_x}{\partial x} + \frac{\partial v_y}{\partial y} \right) \right] \right. \\
&\left. +\frac{1}{2}\frac{\partial}{\partial y} \left[ H \dot{\epsilon}_e^{1/n-1} \left( \frac{\partial v_x}{\partial y} + \frac{\partial v_y}{\partial x} \right) \right] \right) + \tau_{b,x} = \rho g H \frac{\partial h}{\partial x}, \\
&A^{-1/n} \left( \frac{\partial}{\partial y} \left[ H \dot{\epsilon}_e^{1/n-1} \left( 2\frac{\partial v_y}{\partial y} + \frac{\partial v_x}{\partial x} \right) \right] \right. \\
&\left. +\frac{1}{2}\frac{\partial}{\partial x} \left[ H \dot{\epsilon}_e^{1/n-1} \left( \frac{\partial v_y}{\partial x} + \frac{\partial v_x}{\partial y} \right) \right] \right) + \tau_{b,y} = \rho g H \frac{\partial h}{\partial y},
\end{aligned}
\tag{1}
$$

where $v_x$ and $v_y$ are the velocity components in $x$- and $y$-direction, respectively, $H$ is the ice thickness, $h = H + b$ the ice-surface elevation with ice-base elevation $b$ and $n$ denotes Glen's flow-law exponent (Cuffey and Paterson (2010), a common choice is $n = 3$). The effective strain rate $\dot{\epsilon}_e$ (Greve and Blatter, 2009) can be written as

$$
\dot{\epsilon}_e = \left[ \left(\frac{\partial v_x}{\partial x}\right)^2 + \left(\frac{\partial v_y}{\partial y}\right)^2 + \frac{\partial v_x}{\partial x}\frac{\partial v_y}{\partial y} + \frac{1}{4}\left(\frac{\partial v_x}{\partial y} + \frac{\partial v_y}{\partial x}\right)^2 \right]^{1/2},
\tag{2}
$$

We choose the basal shear stress in Eqs. (1), $\boldsymbol{\tau}_b = (\tau_{b,x}, \tau_{b,y})$, to be given by a Weertman-type sliding law (Greve and Blatter, 2009):

$$
\boldsymbol{\tau}_b = -C|\boldsymbol{v}|^{m-1}\boldsymbol{v},
\tag{3}
$$





with horizontal velocity vector $\boldsymbol{v} = (v_x, v_y)$ and constant friction coefficient $C$. The exponent $m$ determines the particular type of the sliding law including plastic ($m = 0$, magnitude of basal shear stress independent of velocity, Tulaczyk et al., 2000) and linear-viscous ($m = 1$, basal shear stress proportional to ice velocity, MacAyeal, 1989) behavior. A value of $m = 1/n = 1/3$ is commonly assumed to represent sliding over rough bed (Schoof, 2007a; Joughin et al., 2009; Cuffey and Paterson, 2010).

The evolution equation for the ice thickness, i.e., the ice thickness equation (ITE), which results out of mass conservation (Greve and Blatter, 2009) reads

$$\frac{\partial H}{\partial t} = -\text{div}\, \boldsymbol{Q} + a, \tag{4}$$

with horizontal ice flux $\boldsymbol{Q} = H\boldsymbol{v}$ and surface mass balance $a$.

## 2.2 Flow-line case

In the flow-line case the geometry of an ice sheet can be scaled in horizontal ($x$) and vertical ($z$) direction, using two scaling factors $\alpha$ and $\beta$, respectively ($\alpha, \beta > 1$ for stretching and $\alpha, \beta < 1$ for compression). We define these as

$$x' = \alpha x, \tag{5}$$
$$h'(x') = H'(x') + b'(x') = \beta H(x) + \beta b(x) = \beta h(x), \tag{6}$$

where the prime denotes the scaled system. In particular, Eq. (5) states that the ice-sheet length $L$ scales according to $L' = \alpha L$.

Since we neglect the $y$-direction here, we only have to consider the $x$-component of the SSA (Eq. 1a) in which all terms that include $y$ drop out. The effective strain rate (Eq. 2) thus simplifies to $\dot{\epsilon}_e = \left| \frac{\partial v_x}{\partial x} \right|$ and the SSA reads

$$2A^{-1/n} \frac{\partial}{\partial x} \left[ H \left| \frac{\partial v_x}{\partial x} \right|^{1/n-1} \frac{\partial v_x}{\partial x} \right] - C v_x^m - \rho g H \frac{\partial(H+b)}{\partial x} = 0. \tag{7}$$

The ITE (Eq. 4) in flow line is given by

$$\frac{\partial H}{\partial t} = -\frac{\partial(H v_x)}{\partial x} + a. \tag{8}$$

Now we bring these two equations into non-dimensionalized form by introducing the dimensionless variables $H^* = \frac{H}{\mathcal{H}}$, $b^* = \frac{b}{\mathcal{H}}$ and $v_x^* = \frac{v_x \mathcal{T}}{\mathcal{L}}$, using the scales $\mathcal{H}$, $\mathcal{L}$ and $\mathcal{T}$ for ice-sheet thickness, length and response time, respectively. We obtain

$$\underbrace{\frac{2A^{-1/n}\mathcal{T}^{-1/n}}{\rho g \mathcal{H}}}_{=\theta} \frac{\partial}{\partial x^*} \left[ H^* \left| \frac{\partial v_x^*}{\partial x^*} \right|^{1/n-1} \frac{\partial v_x^*}{\partial x^*} \right]$$
$$- \underbrace{\frac{C\mathcal{L}^{m+1}\mathcal{T}^{-m}}{\rho g \mathcal{H}^2}}_{=\phi} v_x^{*m} - H^* \frac{\partial(H^* + b^*)}{\partial x^*} = 0, \tag{9}$$

and

$$\frac{\partial H^*}{\partial t^*} = -\frac{\partial(H^* v_x^*)}{\partial x^*} + \underbrace{\frac{a\mathcal{T}}{\mathcal{H}}}_{=\omega}, \tag{10}$$

for the SSA and ITE, respectively. In Eq. (9) the two dimensionless constants $\theta$ and $\phi$ relate the different involved stresses to the driving stress. Extending $\theta$ with $\mathcal{H}/\mathcal{L}$ and $\phi$ with $\mathcal{L}^{-1}$ we see that these scale factors relates the membrane stresses (Hindmarsh, 2006) and the basal stresses to the driving stress, respectively. Here we do not assume one of the limits for which the driving stress is either fully supported by membrane stresses ($\phi \ll \theta$, situation in an ice shelf) or basal shear stresses ($\phi \gg \theta$, holding for ice frozen to bedrock), respectively, but consider the general case in which non of the stress balance terms are neglected.

The two governing equations (9) and (10) of our problem remain exactly the same as long as each of the dimensionless factors $\theta$, $\phi$ and $\omega$ are kept constant. In other words, the ice-sheet dynamics are expected to be similar under a transformation that leaves these factors unchanged. Thus the scaling of the ice sheet's typical length and thickness scales according to Eqs. (5) and (6), i.e., $\mathcal{L}' = \alpha \mathcal{L}$ and $\mathcal{H}' = \beta \mathcal{H}$ in general requires (some of) the physical parameters $a$, $C$, $A$ and its response time $\mathcal{T}$ to change in order to maintain similarity with respect to the unscaled ice sheet. We hence can infer three scaling conditions for the time-scale ratio $\tau = \mathcal{T}'/\mathcal{T}$:

$$\phi' = \phi \quad \Rightarrow \quad \tau = \alpha^{1+1/m}\beta^{-2/m}\gamma^{1/m}, \tag{11}$$
$$\theta' = \theta \quad \Rightarrow \quad \tau = \beta^{-n}\zeta^{-1}, \tag{12}$$
$$\omega' = \omega \quad \Rightarrow \quad \tau = \beta\delta^{-1}, \tag{13}$$

with friction-coefficient ratio $\gamma = C'/C$, ice-softness ratio $\zeta = A'/A$ and surface-mass-balance ratio $\delta = a'/a$. This system of 3 equations has 6 unknowns from which 4 remain when we take $\alpha$ and $\beta$ as given by the applied geometric transformation. Prescribing one of the three parameter ratios $\gamma$, $\delta$ or $\zeta$ hence determines the scaling of the other two parameters and the time scaling of the system.

We can link the ratios of surface mass balance and ice softness by combining Eqs. (12) and (13), yielding

$$\delta = \beta^{n+1}\zeta, \tag{14}$$

a relation which is independent of the horizontal scaling factor $\alpha$. For the case of a scaled ice-sheet geometry that is left unchanged in vertical direction ($\beta = 1$) ice softness and accumulation hence scale identically.

Using Eqs. (11)-(13) we can further express $\delta$ and $\zeta$ as functions of both geometric scaling ratios and the basal friction ratio:

$$\delta = \alpha^{-(1+1/m)}\beta^{1+2/m}\gamma^{-1/m}. \tag{15}$$
$$\zeta = \alpha^{-(1+1/m)}\beta^{-n+2/m}\gamma^{-1/m}. \tag{16}$$



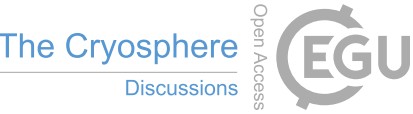

Inserting Eq. (14) into Eq. (16) we also obtain a condition for the basal-friction ratio as a function of both geometric scaling parameters and the surface-mass-balance ratio:

$$\gamma = \alpha^{-(1+m)}\beta^{2+m}\delta^{-m}. \tag{17}$$

Results of an application of the derived scaling laws in numerical flow-line simulations are given in Sec. 3

### 2.3 Consistency with flow-line boundary-layer theory

Here we show that the scaling conditions derived above by dimensional analysis under the concept of similitude, are consistent with the boundary-layer theory which was introduced by Schoof (2007b) for an unbuttressed, isothermal, flow-line ice sheet in SSA. Neglecting membrane stresses in the stress balance, matched asymptotics are applied to solve a boundary-layer problem for the transition zone between grounded and floating ice. The ice-sheet surface slope is then given by (Schoof, 2007b, Eq. 25)

$$\frac{\partial h(x)}{\partial x} = \frac{\partial(H(x)+b(x))}{\partial x} = \frac{C}{\rho g}\frac{|Q(x_{gl})|^{m-1}Q(x_{gl})}{h(x)^{m+1}}, \tag{18}$$

where $x_{gl}$ denotes the grounding-line position and $Q(x_{gl})$ is the flux across the grounding line. According to Eqs. (5) and (6) the scaling of the surface slope reads

$$\frac{\partial h'(x')}{\partial x'} = \frac{\beta}{\alpha}\frac{\partial h(x)}{\partial x} \tag{19}$$

and in combination with Eq. (18) we can write

$$\frac{C'}{\rho g}\frac{\left|Q'(x'_{gl})\right|^{m-1}Q'(x'_{gl})}{h'(x')^{m+1}} = \frac{\beta}{\alpha}\frac{C}{\rho g}\frac{|Q(x_{gl})|^{m-1}Q'(x_{gl})}{h(x)^{m+1}}. \tag{20}$$

Presuming that the flux across the grounding line is always positive in $x$-direction and using once again Eq. (6) yields a scaling relation for the grounding-line flux

$$Q'(x'_{gl}) = \alpha^{-1/m}\beta^{(1+2/m)}\gamma^{-1/m}Q(x_{gl}). \tag{21}$$

The boundary-layer method considers the ITE in steady state ($\frac{\partial H}{\partial t}=0$) and hence integration of (Eq. 4) over the entire ice-sheet length yields

$$Q(x_{gl}) = aL. \tag{22}$$

Inserting this expression for the grounding-line flux into Eq. (21) we arrive at the same condition for the scaling of the surface mass balance (Eq. 15) that we obtained from principle of similitude in the previous section.

A central result of the boundary layer theory is an analytic solution for the grounding-line flux as a function of ice thickness at the grounding line (Schoof2007, Eq. (16)):

$$Q(x_{gl}) = \left(\frac{A(\rho g)^{1+n}(1-\rho/\rho_w)^n}{4^nC}\right)^{\frac{1}{m+1}}H(x_{gl})^{\frac{m+n+3}{m+1}}. \tag{23}$$

Inserting this relation into Eq. (21) and applying some basic algebra we obtain the same scaling relation for the ice softness as derived in the previous sections (Eq. 16).

Setting $Q(x_{gl}) = v_x(x_{gl})H(x_{gl})$ in Eq. (22) and dividing by $Q'(x_{gl})$ we obtain

$$\frac{v'_x(x'_{gl})}{v_x(x_{gl})} = \alpha\beta^{-1}\delta. \tag{24}$$

Since the boundary-layer theory assumes steady-state conditions, we introduce a velocity scale $\mathcal{V} = \mathcal{L}/\mathcal{T}$ to be able to derive a response-time relation. This yields $v'_x/v_x = \alpha/\tau$ and the response-time scaling law resulting from Eq. (24) is identical to Eq. (13).

Thus the same 3 independent equations that determine the ice-sheet scaling behavior and were derived by the means of similarity analysis in the previous section also result from boundary-layer theory.

### 2.4 Two-dimensional case with one time and one length scale

The two-dimensional SSA (Eq. 1) is derived from the full-Stokes equation using a single horizontal length scale $\mathcal{L}$ and time scale $\mathcal{T}$, respectively (Greve and Blatter, 2009). Continuing this line of thought, we introduce the dimensionless velocity in $y$-direction, $v_y^* = \frac{v_y\mathcal{T}}{\mathcal{L}}$, in addition to the dimensionless variables from Sec. 2.2 to non-dimensionalize the SSA equations. The dimensionless effective strain rate (Eq. 2) then reads

$$\dot{\epsilon}_e^* = \mathcal{T}\dot{\epsilon}_e \tag{25}$$

For the $x$-component of the SSA (Eq. 1a) we hence obtain

$$\underbrace{\frac{A^{-1/n}\mathcal{T}^{-1/n}}{\rho g\mathcal{H}}}_{=\Theta}\left(\frac{\partial}{\partial x^*}\left[\dot{\epsilon}_e^{*1/n-1}H^*\left(2\frac{\partial v_x^*}{\partial x^*}+\frac{\partial v_y^*}{\partial y^*}\right)\right]\right.$$
$$+\frac{1}{2}\frac{\partial}{\partial y^*}\left[\dot{\epsilon}_e^{*(1-n)/n}H^*\left(\frac{\partial v_x^*}{\partial y^*}+\frac{\partial v_y^*}{\partial x^*}\right)\right]\right)$$
$$-\underbrace{\frac{C\mathcal{L}^{m+1}\mathcal{T}^{-m}}{\rho g\mathcal{H}^2}}_{=\Phi}v_x^{*m}-H^*\frac{\partial(h^*+b^*)}{\partial x^*}=0. \tag{26}$$

The same coefficients $\Theta$ and $\Phi$ result from the $y$-component of the SSA, which is not specified here. The non-dimensionalized ITE (Eq. 4) reads

$$\frac{\partial H^*}{\partial t^*} = -\mathrm{div}\left(H^*\boldsymbol{v}^*\right)+\underbrace{\frac{a\mathcal{T}}{\mathcal{H}}}_{=\Omega}. \tag{27}$$

Comparison between the flow-line and the two-dimensional SSA and ITE shows that we obtained the same number of



dimensionless factors that appear at the same place and are identical to each other, i.e., $\theta = \Theta$, $\phi = \Phi$ and $\omega = \Omega$. Hence under the assumption of a single horizontal length scale the scaling relations for the two-dimensional SSA are the same as in the flow-line case.

### 2.5 Two-dimensional case with time and length scales for both horizontal directions

Starting again from the two-dimensional SSA (Eq. 1) we now make the less-constraining assumption of two horizontal length scales $\mathcal{L}_x$ and $\mathcal{L}_y$ and accordingly two time scales $\mathcal{T}_x$ and $\mathcal{T}_y$, yielding the dimensionless velocities $v_x^* = \frac{v_x \mathcal{T}_x}{\mathcal{L}_x}$ and $v_y^* = \frac{v_y \mathcal{T}_y}{\mathcal{L}_y}$. In this case the effective strain rate (Eq. 2) does not simplify to a single term as in the previous sections but consists of several mixed terms. The SSA thus expands to a much longer expression which we detail in the Appendix A. Although we obtain a multiple of dimensionless coefficients that need to remain constant for the ice sheet to fulfill similarity under scaling, the resulting scaling laws are identical to the ones derived above (see Appendix A). This implies that our requirement of similarity results in the constraint that the ice sheet can have only one time scale $\mathcal{T} = \mathcal{T}_x = \mathcal{T}_y$ and one length scale $\mathcal{L} = \mathcal{L}_x = \mathcal{L}_y$ as opposed to our initial assumption of distinct scales for each horizontal direction.

We investigate ice-sheet scaling also in a three-dimensional setup in the next section.

### 3 Comparison with simulations

We compare our analytical findings with results from numerical simulations applying the Parallel Ice Sheet Model in conceptual geometric setups. The model is the same as used in (Feldmann and Levermann, 2015a) but here run in SSA-only mode. We define a reference topographic geometry which is prescribed in an unscaled reference experiment (indexed as "ref") along with the parameter values shown in Table 1. The scaling experiments use geometrically scaled versions of the reference bed topography and the physical parameters are modified according to the scaling laws derived in Sec. 2.2.

Halfing the horizontal and/or vertical length scales of the reference topography we obtain three geometric configurations which are shrinked in vertical $(\alpha, \beta) = (1.0, 0.5)$, horizontal $(\alpha, \beta) = (0.5, 1.0)$ or both directions $(\alpha, \beta) = (0.5, 0.5)$, respectively. To be able to calculate the other physical parameters $a, A, C$ that apply to the scaling experiments according to the 3 scaling relations (Eqs. 15 - 17) we need to prescribe one more scaling ratio in addition to $\alpha$ and $\beta$. Setting $\gamma = 1$ (constant basal friction) and $\delta = 1$ (constant surface mass balance), thus two sub-sets of simulations are generated. The resulting scaling ratios which determine the parameter values are shown in Table ?? for each of the seven experiments. We apply the described procedure using 1) a flow-line setup (one horizontal and one vertical direction, bed topography in black in Fig. 2) and 2) a three-dimensional channel-flow setup (flow-line setup extended by second horizontal direction, bed topography shown in Figs. 3 and 4) as detailed in Appendix B.

The experiments are designed to perturb an ice sheet in equilibrium, triggering a marine ice-sheet instability that unfolds unaffected by the ceased perturbation. The speed of unstable grounding-line retreat and the equilibrium ice-sheet profiles before and after the instability serve as a measure to compare the scaling of the dynamic response and the steady-state geometry, respectively.

### 3.1 Comparing time scales of instability

All of our simulations show a similar pattern of grounding-line evolution after perturbation (Figs. 5 and 6): After a phase of little to negligible grounding-line retreat the retreat rate increases (grounding line passes the coastal sill and enters the retrograde slope), reaching its maximum value around the minimum of the bed depression before declining to zero (grounding line stabilizes on inland up-sloping bed). The initial and final grounding-line positions of comparable setups (continuous lines) match or are close to each other. The similarity of ice-sheet shapes between different geometric configurations becomes apparent when laying the modeled steady-state ice-sheet profiles on top of each other and scaling the spatial axes according to $\alpha$ and $\beta$ (shown exemplarily in Figs. 2 and 3).

The simulations clearly differ in the time scale of the MISI evolution which can be measured by the grounding-line retreat rate $\dot{x}_{gl} = \frac{\partial x_{gl}}{\partial t}$. To compare different simulations we introduce a retreat-rate scaling ratio:

$$\dot{\chi} = \frac{\dot{x}_{gl}'}{\dot{x}_{gl}} = \frac{\alpha}{\tau}. \tag{28}$$

Dependent on which additional parameter is held constant under geometric scaling, we replace the time-scale ratio using Eq. (11) or Eq. (13) to obtain scaling laws for the retreat rate as functions of the geometric scaling ratios only:

$$\gamma = 1 \quad \Rightarrow \quad \dot{\chi} = \alpha^{-1/m} \beta^{2/m}, \tag{29}$$
$$\delta = 1 \quad \Rightarrow \quad \dot{\chi} = \alpha \beta^{-1}. \tag{30}$$

We can thus calculate the retreat-rate ratios for all considered geometric configurations (Table ??). The grounding-line curves of our simulations are approximately linear over the time period during which the grounding line passes the bed depression and its retreat rate is largest. We fit a slope to the linear section of the unscaled simulation (purple slope fitted to black curve in Figs. 5 and 6), to obtain our reference retreat rate. Using the calculated retreat-rate ratios from Table ?? we can predict the grounding-line retreat rates for the scaled setups. Superimposing the linear sections of the scaled experiments with the respective analytically calculated slope (Figs. 5 and 6) gives a good match between numerical results





and theory. Our simulation ensemble of scaled ice sheets thus exhibits similarity as predicted from theory, regarding transient ice-sheet dynamics and steady-state geometry.

## 4   Implications for the response times of ice sheets

Based on the scaling laws derived in Sec. 2 we explore analytically the implications of a scaling of ice-sheet parameters and geometry for the response-time scaling. Making the assumption of a constant basal friction parameter ($\gamma = 1$) while allowing a variation in surface mass balance and ice softness we are able to calculate the response-time ratio $\tau$ (Eq. 11) as a function that only depends on the geometric scaling ($\alpha$ and $\beta$) and the friction exponent $m$:

$$\tau = \alpha^{1+1/m}\beta^{-2/m}. \tag{31}$$

Using this equation in combination with Eqs. (12) and (13) we obtain contour maps for the ratios $\tau$, $\zeta$ and $\delta$ in the $\alpha$-$\beta$ phase space (Figs. 7a-c for the common choice of an exponent value of $m = 1/n$ with $n = 3$ (Schoof, 2007a; Greve and Blatter, 2009; Cuffey and Paterson, 2010). Therein the blue and red areas correspond to the regimes of an increasing and decreasing parameter value under geometric scaling, respectively, which are separated by a white curve along which the considered parameter remains constant.

### 4.1   Linking horizontal and vertical scales

To be able to follow physically motivated curves through the phase space we link the horizontal and the vertical scale. Motivated by the Vialov ice-sheet profile (Vialov, 1958; Greve and Blatter, 2009), for which the central (maximum) ice-sheet thickness is proportional to the square root of the ice-sheet length we assume a relation between the ice-thickness scale $\mathcal{H}$ and the length scale $\mathcal{L}$ of the form

$$\mathcal{H} \sim \mathcal{L}^q \quad \text{with} \quad 0 < q \le 1. \tag{32}$$

With $\alpha = \mathcal{L}'/\mathcal{L}$ and $\beta = \mathcal{H}'/\mathcal{H}$ it follows that for the postulated ice-sheet proportion the two geometric scaling factors are linked such that

$$\beta = \alpha^q, \tag{33}$$

and Eq. (31) then reads

$$\tau = \alpha^{\frac{m-2q+1}{m}}. \tag{34}$$

We are interested in finding a critical value of the exponent in Eq. (31) which determines a threshold in the $\alpha$-$\beta$ phase space between the two regimes of increasing ($\tau < 1$) and decreasing ($\tau > 1$) ice-sheet response time under applied geometric scaling. Assuming horizontal stretching ($\alpha > 1$), which according to Eq. (33) implies also vertical stretching ($\beta > 1$,

see Fig. 7d), it follows that $\tau < 1$ only if the exponent in Eq. (34) is negative. Hence there exists a critical threshold

$$q_c = \frac{m+1}{2}, \tag{35}$$

with $m \in (0,1]$ and thus $q_c \in (\frac{1}{2},1]$, above which the scaled, i.e. stretched, system responds faster compared to the unscaled system. This is visualized in Fig. 7a for $m = 1/3$. The area between the dashed ($q = q_c = 2/3$) and the continuous ($q = 1$) curves is in the regime of $\tau < 1$ for $\alpha > 1$. Vice versa, for a shrinked ice sheet ($\alpha < 1$ and hence $\beta < 1$) in the area between these two curves holds $\tau > 1$. The same qualitative scaling applies to the ice softness whereas the surface mass balance scales oppositely (Figs. 7b and c).

An exponent of $q = 1/2$ which represents Vialov proportions constitutes the lower aymptotic limit of the domain of all possible $q_c$ (limit $m \to 0$, Eq. 34 requires $m > 0$ for $\alpha$ to remain finite). Thus a Vialov-shaped ice sheet exhibits a response-time scaling oppositely to the scaling explained above (the dotted Vialov curve in Fig. 7a lies always outside the region between continuous and dahed curve, independently of $m$).

Assuming Vialov conditions under constant friction, the scaling of the response time (Eq. 31), surface mass balance (Eq. 15) and ice softness (Eq. 16), respectively, becomes independent of $m$ which is visualized in Fig 8. Evaluating the curves in the left vicinity of $\alpha = 1$, meaning a small reduction in both vertical and horizontal ice-sheet extent, yields a plausible scaling of the ice-sheet parameters in a warming atmosphere: Rising atmospheric temperatures cause an increase in surface mass balance ($\delta > 1$ in Fig. 7c, Frieler et al., 2015) and also lead to a softening of the ice ($\zeta > 1$ in Fig. 7b, Cuffey and Paterson, 2010). The response time then decreases ($\tau < 1$ in Fig. 7a). In this picture a warming-induced ice-sheet retreat would hence shift the ice sheet into the regime of faster response to perturbation, tending to accelerate potential further retreat.

### 4.2   Role of basal friction exponent $m$

The response-time scaling considered here is a function of the basal friction exponent $m$ (Eq. 31) and the visualization of the response-time ratio in the $\alpha$-$\beta$ phase space (Fig. 7 accounts for only one value of $m$. To examine the influence of $m$ on the scaling we cut several hypersurfaces through the phase space, sampling the domain of the exponent.

Fixing the horizontal scale, i.e., going along $\alpha = 1$, yields that vertical stretching (shrinking) always results in a short (longer) ice-sheet response time (Fig. 9a). In this case the parameter choice of $m$ only determines the curvature of $\tau(\beta)$. Fixing the vertical scale ($\beta = 1$) results in opposite behavior of $\tau$, i.e., horizontal stretching (shrinking) always yields a longer (shorter) ice-sheet response time (Fig. 9b). Equal geometric scaling of the two directions ($\alpha = \beta$) gives a similiar picture as obtained for $\alpha = 1$ (the magnitude of the negative



$\beta$-exponent is always larger than the $\alpha$-exponent), with the difference that here the time scaling becomes independent of the geometric scaling for $m = 1$ (Fig. 9c).

Requiring the response-time scaling law (Eq. 31) to be independent of $m$ yields the relation $\beta = \alpha^{\frac{1-m(k-1)}{2}}$ (with $k$ a real number) and thus $\tau = \alpha^k$. In general, a negative (positive) value of $k$ then results in a faster (slower) response when stretching (shrinking) the ice sheet horizontally. The case of $k = 0$ yields a constant time scale ($\tau = 1$), independent of the $\alpha$ value (Fig. 9d). The case of $k = 1$ corresponds to the Vialov case for which the time-scale ratio increases linearly when stretching the ice sheet horizontally.

## 5 Discussion and conclusions

Carrying out a dimensional analysis of the stress balance in SSA and the equation of mass conservation we derive ice-sheet scaling conditions for the vertical and horizontal length scales, the response time and the relevant physical parameters which determine ice-sheet behavior.

Specifically, we find that the scaling relations derived for the SSA in flow line (Eqs. 11-13) also hold for the SSA in two horizontal dimensions under the assumtion of a single horizontal time and length scale, respectively. Only the two-dimensional SSA accounts for stress components that allow for horizontal shearing and hence the effect of buttressing.

Our analysis also shows that although the full SSA accounts for both horizontal dimensions there can only exist one time scale $\mathcal{T}$ and one length scale $\mathcal{L}$, as opposed to one for each dimension ($\mathcal{T}_x$, $\mathcal{T}_y$ and $\mathcal{L}_x$, $\mathcal{L}_y$) under the principle of similitude. To non-dimensionalize the stress balance we introduce scales for ice-sheet length, thickness and time without assuming typical numerical values for these scales. We thus do not compare orders of magnitudes of acting stresses to neglect terms in the stress balance as is often done in the course of a dimensional analysis (citation) but consider the general case of comparable magnitudes of membrane and basal stresses, respectively. In other studies not only ice-sheet length, thickness and velocity but also the friction parameter $C$ is expressed by typical scales of length, thickness and time resulting in a dimensionless stress balance that is characterized by a single scaling parameter (often denoted as $\epsilon$, Schoof, 2007a; Tsai et al., 2015). In the present study we consider $C$ as an independent parameter and thus obtain two scaling parameters $\theta$ and $\phi$ in the stress balance (Eqs. 9 and 26). The resulting scaling laws hence involve the scaling of the basal roughness explicitly. The same holds for the scaling of the surface mass balance $a$.

The scaling laws derived here are consistent with boundary-layer theory which considers the transition zone between the grounded and floating regimes of a rapidly sliding equilibrium ice sheet in flow line (Schoof, 2007b). The conditions that don't involve a time scale (Eqs. 15 and 16) follow directly out of the analytic equations for steady-state ice-sheet geometry and the grounding-line flux that result from boundary-layer theory. To obtain the scaling relation also for the ice-sheet response time (Eq. 13) out of the steady-state theory it is necessary to introduce a velocity scale.

The presented scaling conditions can provide rules in the design of model setups for numerical simulations to obtain parameter sets that leave the ice-sheet geometry (absolute shape and extent) unchanged. For instance, a doubling of the basal-friction parameter under constant surface mass balance requires the ice softness to be reduced to 1/8, or a doubling in surface mass balance under constant basal friction requires a doubling of the ice-softness value.

For the numerical simulations conducted in this study we apply parameter configurations that half the geometric scale in horizontal and/or vertical direction with respect to the reference. The resulting ice-sheet response times range over three orders of magnitude (see Table **??**). Irrespective of whether in a two- or three-dimensional setup the modeled ice-sheet dynamics, represented by the rate of unstable grounding-line retreat (Figs. 5 and 6) as well as the geometry, represented by ice-sheet shape and grounding line position in equilibrium (Figs. 2 – 4), exhibit the scaling behavior predicted from the analytical calculations to a good approximation. For the flow-line setup three scaled parameter sets show different qualitative ice-sheet evolution compared to the reference, while still complying with the expected response-time scaling. This difference is attributed to the design of the reference setup, i.e., the closeness of the initial steady-state grounding line to the point of instability (local bed maximum). Very small deviations from this position trigger unperturbed instability or prevent landward induced instability in the scaled setups (see Appendix B).

In contrast to the flow-line configuration the three-dimensional setup inherently accounts for the buttressing effect in the initial steady-state simulation due to the presence of a confined ice shelf (Dupont and Alley, 2005; Gudmundsson et al., 2012). However, the ice shelf is removed in the course of perturbation to prevent scale-dependent influences that would originate from a forcing through sub-shelf melting, surface accumulation or ice softness. Thus the speed of grounding line retreat (and hence ice-sheet response time) is only indirectly affected by the former buttressing effect. An investigation of the response-time scaling under direct influence of ice-shelf buttressing requires a carefully designed experimental setup that maintains the ice shelf during perturbation (as in Asay-Davis et al., 2015) and accounts for the scaling also in the applied forcing.

To analytically investigate the implications of geometric scaling for the ice-sheet response time we make the simplifying assumption of constant basal friction ($\gamma = 1$). Though the response-time scaling law still then still depends on the sliding exponent $m$ (Eq. 31) the qualitative response-time scaling (shorter or longer response time) turns out to be independent of the choice of $m$ (9): Vertical ice-sheet stretching (compression) leads to a faster (slower) ice-sheet response



and the opposite holds for the horizontal direction. In other words, thicker or shorter ice sheets tend to respond faster than thinner or longer ones. Equal scaling in horizontal and vertical direction ($\alpha = \beta$) yields that larger ice sheets respond faster than smaller ones.

Assuming a relation between the horizontal and vertical scale of the form $\beta = \alpha^q$ with $0 < q \le 1$, we find a critical $m$-dependent threshold $q_c$ for the exponent (Eq. 35) above which larger (smaller) ice sheets always exhibit a shorter (longer) response time. The case of $q = 1/2$ represents the lower asymptotic limit for all possible $q_c$ and corresponds to an ice sheet with Vialov-type proportions for which the central ice thickness is the square root of the horizontal extent. Conceptual flow-line experiments similar to the ones conducted here (Feldmann and Levermann, 2015a) revealed that the Vialov profile, which results under simplified conditions from the shallow-ice approximation of the full-Stokes stress balance in flow line, can also reasonable approximate the ice-sheet shape in SSA. In the same study a comparison between steady-state ice-sheet profiles before and after collapse suggested a scaling of $\beta = \alpha^{1/2}$. For such an ice sheet the time scaling is identical to the scaling of its length, i.e., stretching (compression) results in slower (faster) response which is opposite behavior than for the above discussed case of $q > q_c > 1/2$. A thought experiment that is consistent with the scaling behavior derived for this kind of profile reveals that in the course of an ice-sheet retreat that is triggered by atmospheric warming the ice-sheet response would become faster, with self-accelerating effect on further retreat (Fig 8). Note that all the consideration made above are only valid for a constant basal-friction parameter.

In place of prescribing basal friction, the assumption of a constant surface mass balance ($\delta = 1$) or ice softness ($\zeta = 1$), results in a more trivial response-time scaling which either equals the vertical scaling (Eq. 13) or depends on the vertical scaling via a power-law relation with exponent $-n$ (Eq. 12), respectively. Since $n$ is always positive (often chosen as 3, see Cuffey and Paterson, 2010) also here the qualitative time scaling does not depend on the parameter value. There are several other ways to analyze the implications of the scaling conditions derived here on ice-sheet dynamics that are not covered in this study.

Our approach includes several assumptions (shallow stress balance, isothermal ice flow, choice of sliding law, parameter constraints) and thus simplifies the problem of ice sheet flow. At the same time it allows for the fundamental scaling analysis conducted here which incorporates the relevant physics of fast ice flow and results in scaling conditions that relate important physical parameters of an ice sheet to each other.

The applied Weertman-type sliding law (Eq. 3) is a common choice (Fowler, 1981; Schoof, 2007a; Pattyn et al., 2013) amongst others used to describe the sliding of ice sheets over bedrock (Greve and Blatter, 2009; Cuffey and Paterson, 2010; Tsai et al., 2015). It covers diverse types of sliding behavior depending on the sliding exponent $m$ in Eq. (31). Except for the plastic limit ($m = 0$) it relates the scale of basal stress to the scale of velocity, resulting in a scaling law which links the scaling of ice-sheet geometry, friction and response time, respectively (Eq. 11).

Our analytic exploration of the derived ice-sheet scaling behavior applies several constraints to the parameter space and is thus far from being holistic but is aimed to allow for (simplified) statements on the influence of geometric scaling on response time. The set of scaling conditions presented here shall provide a model which allows for a fundamental comparison of the large-scale scaling of the geometry and relevant parameters that determine ice-sheet dynamics. In particular the response-time scaling conditions might be suitable to analyze speed of the transient response to climatic perturbations of the polar ice sheets that took place in the past or might become relevant for the future.

## Appendix A: Two-dimensional case with two time and length scales for both horizontal directions

Introducing the dimensionless velocities $v_x^* = \frac{v_x \mathcal{T}_x}{\mathcal{L}_x}$ and $v_y^* = \frac{v_y \mathcal{T}_y}{\mathcal{L}_y}$ the non-dimensionalized form of the effective strain rate (Eq. 2) reads

$$
\dot\epsilon_e = \left[ \mathcal{T}_x^{-2} \left( \frac{\partial v_x^*}{\partial x^*} \right)^2 + \mathcal{T}_y^{-2} \left( \frac{\partial v_y^*}{\partial y^*} \right)^2 + \mathcal{T}_x^{-1} \mathcal{T}_y^{-1} \frac{\partial v_x^*}{\partial x^*} \frac{\partial v_y^*}{\partial y^*} \right.
$$
$$
\left. + \frac{1}{4} \left( \mathcal{L}_x \mathcal{L}_y^{-1} \mathcal{T}_x^{-1} \frac{\partial v_x^*}{\partial y^*} + \mathcal{L}_x^{-1} \mathcal{L}_y \mathcal{T}_y^{-1} \frac{\partial v_y^*}{\partial x^*} \right)^2 \right]^{1/2}.
$$

$$(A1)$$





Insertion into Eq. (1a) yields the following expression for the $x$-component of the two-dimensional SSA:

expression $I$ reads:

$$
\begin{aligned}
&\frac{\partial}{\partial x^*}\left[H^*\left(\underbrace{\left[\frac{2A^{-1/n}}{\rho g \mathcal{H}\mathcal{T}_x}\right]^{\frac{2n}{1-n}}\dot{\epsilon}_e^2}_{I}\right)^{\frac{1-n}{2n}}\frac{\partial v_x^*}{\partial x^*}\right.\\
&\left.+H^*\left(\underbrace{\left[\frac{A^{-1/n}}{\rho g \mathcal{H}\mathcal{T}_y}\right]^{\frac{2n}{1-n}}\dot{\epsilon}_e^2}_{II}\right)^{\frac{1-n}{2n}}\frac{\partial v_y^*}{\partial y^*}\right]\\
&+\frac{1}{2}\frac{\partial}{\partial y^*}\left[H^*\left(\underbrace{\left[\frac{A^{-1/n}\mathcal{L}_x^2}{\rho g \mathcal{H}\mathcal{L}_y^2\mathcal{T}_x}\right]^{\frac{2n}{1-n}}\dot{\epsilon}_e^2}_{III}\right)^{\frac{1-n}{2n}}\frac{\partial v_x^*}{\partial y^*}\right.\\
&\left.+H^*\left(\underbrace{\left[\frac{A^{-1/n}}{\rho g \mathcal{H}\mathcal{T}_y}\right]^{\frac{2n}{1-n}}\dot{\epsilon}_e^2}_{IV}\right)^{\frac{1-n}{2n}}\frac{\partial v_y^*}{\partial x^*}\right]\\
&+\underbrace{\frac{\mathcal{L}_x^{m+1}C}{\rho g \mathcal{H}^2\mathcal{T}_x^m}}_{=\Phi_x}v_x^{*m}-H^*\frac{\partial(h^*+b^*)}{\partial x^*}=0,
\end{aligned}
\tag{A2}
$$

$$
\begin{aligned}
I=&\underbrace{\left[\frac{2A^{-1/n}}{\rho g \mathcal{H}\mathcal{T}_x^{1/n}}\right]^{\frac{2n}{1-n}}}_{\Theta_{I,1}}\left(\frac{\partial v_x^*}{\partial x^*}\right)^2\\
&+\underbrace{\left[\frac{2A^{-1/n}}{\rho g \mathcal{H}\mathcal{T}_x\mathcal{T}_y^{\frac{1-n}{n}}}\right]^{\frac{2n}{1-n}}}_{\Theta_{I,2}}\left(\frac{\partial v_y^*}{\partial y^*}\right)^2\\
&+\underbrace{\left[\frac{2A^{-1/n}}{\rho g \mathcal{H}\mathcal{T}_x^{\frac{1+n}{2n}}\mathcal{T}_y^{\frac{1-n}{2n}}}\right]^{\frac{2n}{1-n}}}_{\Theta_{I,3}}\frac{\partial v_x^*}{\partial x^*}\frac{\partial v_y^*}{\partial y^*}\\
&+\frac{1}{4}\left(\underbrace{\left[\frac{2A^{-1/n}}{\rho g \mathcal{H}\mathcal{T}_x^{1/n}}\left(\frac{\mathcal{L}_x}{\mathcal{L}_y}\right)^{\frac{1-n}{n}}\right]^{\frac{n}{1-n}}}_{\Theta_{I,4}}\frac{\partial v_x^*}{\partial y^*}\right.\\
&\left.+\underbrace{\left[\frac{2A^{-1/n}}{\rho g \mathcal{H}\mathcal{T}_x\mathcal{T}_y^{\frac{1-n}{n}}}\left(\frac{\mathcal{L}_y}{\mathcal{L}_x}\right)^{\frac{1-n}{n}}\right]^{\frac{n}{1-n}}}_{\Theta_{I,5}}\frac{\partial v_y^*}{\partial x^*}\right)^2,
\end{aligned}
\tag{A3}
$$

with the dimensionless coefficient $\Phi_x$ which has the same form as $\Phi$ (Eq. 26) but is specific for the $x$-direction. The terms $I$, $II = IV$ and $III$ are evaluated in the following to obtain dimensionless factors for the SSA equation. The first

from which we obtain five dimensionless factors $\Theta_{I,1},...,\Theta_{I,5}$. Applying the same steps for expressions





$II$ and $III$ yields ten more coefficients:

$$II = IV: \quad \Theta_{II,1} = \Theta_{IV,1} = \frac{2A^{-1/n}}{\rho g \mathcal{H} \mathcal{T}_x^{\frac{1-n}{n}} \mathcal{T}_y} \tag{A4}$$

$$\Theta_{II,2} = \Theta_{IV,2} = \frac{2A^{-1/n}}{\rho g \mathcal{H} \mathcal{T}_y^{1/n}} \tag{A5}$$

$$\Theta_{II,3} = \Theta_{IV,3} = \frac{2A^{-1/n}}{\rho g \mathcal{H} \mathcal{T}_x^{\frac{1-n}{2n}} \mathcal{T}_y^{\frac{1+n}{2n}}} \tag{A6}$$

$$\Theta_{II,4} = \Theta_{IV,4} = \frac{2A^{-1/n}}{\rho g \mathcal{H} \mathcal{T}_x^{\frac{1-n}{n}} \mathcal{T}_y} \left( \frac{\mathcal{L}_x}{\mathcal{L}_y} \right)^{\frac{1-n}{n}} \tag{A7}$$

$$\Theta_{II,5} = \Theta_{IV,5} = \frac{2A^{-1/n}}{\rho g \mathcal{H} \mathcal{T}_y^{1/n}} \left( \frac{\mathcal{L}_y}{\mathcal{L}_x} \right)^{\frac{1-n}{n}} \tag{A8}$$

$$III: \quad \Theta_{III,1} = \frac{2A^{-1/n}}{\rho g \mathcal{H} \mathcal{T}_x^{1/n}} \left( \frac{\mathcal{L}_x}{\mathcal{L}_y} \right)^2 \tag{A9}$$

$$\Theta_{III,2} = \frac{2A^{-1/n}}{\rho g \mathcal{H} \mathcal{T}_x \mathcal{T}_y^{\frac{1-n}{n}}} \left( \frac{\mathcal{L}_x}{\mathcal{L}_y} \right)^2 \tag{A10}$$

$$\Theta_{III,3} = \frac{2A^{-1/n}}{\rho g \mathcal{H} \mathcal{T}_x^{\frac{1+n}{2n}} \mathcal{T}_y^{\frac{1-n}{2n}}} \left( \frac{\mathcal{L}_x}{\mathcal{L}_y} \right)^2 \tag{A11}$$

$$\Theta_{III,4} = \frac{2A^{-1/n}}{\rho g \mathcal{H} \mathcal{T}_x^{1/n}} \left( \frac{\mathcal{L}_x}{\mathcal{L}_y} \right)^{\frac{1+n}{n}} \tag{A12}$$

$$\Theta_{III,5} = \frac{2A^{-1/n}}{\rho g \mathcal{H} \mathcal{T}_x \mathcal{T}_y^{\frac{1-n}{n}}} \left( \frac{\mathcal{L}_x}{\mathcal{L}_y} \right)^{\frac{-1+3n}{n}} \tag{A13}$$

In order to obtain the same equations independent of an applied ice-sheet scaling the dimensionless coefficients need to remain constant. We start with the first set of coefficients:

$$\Theta'_{I,1} \overset{!}{=} \Theta_{I,1} \quad \Rightarrow \quad \tau_x = \beta^{-n} \zeta^{-1}, \tag{A14}$$

$$\Theta'_{I,2} \overset{!}{=} \Theta_{I,2} \quad \Rightarrow \quad \tau_x = \beta^{-1} \zeta^{-1/n} \tau_y^{-\frac{1-n}{n}}, \tag{A15}$$

$$\Theta'_{I,3} \overset{!}{=} \Theta_{I,3} \quad \Rightarrow \quad \tau_x = \beta^{-\frac{2n}{1+n}} \zeta^{-\frac{2}{1+n}} \tau_y^{-\frac{1-n}{1+n}}, \tag{A16}$$

$$\Theta'_{I,4} \overset{!}{=} \Theta_{I,4} \quad \Rightarrow \quad \tau_x = \beta^{-n} \zeta^{-1} \left( \frac{\alpha_x}{\alpha_y} \right)^{1-n}, \tag{A17}$$

$$\Theta'_{I,5} \overset{!}{=} \Theta_{I,5} \quad \Rightarrow \quad \tau_x = \beta^{-1} \zeta^{-1/n} \left( \frac{\alpha_x}{\alpha_y} \right)^{\frac{1-n}{n}} \tau_y^{-\frac{1-n}{n}}. \tag{A18}$$

We immediately see that Eq. (A14) gives the same time scaling (in $x$-direction) as derived for the more constraint cases, i.e., in flow-line (Eq. 12) as well as for the two-dimensional case that assumes only one time and length scale, respectively (Sec. 2.4). Comparison of Eqs. (A14) and (A17) directly yields the condition $\alpha_x = \alpha_y$. Furthermore, replacing $\beta$ and $\zeta$ in Eq. (A15) using Eq. (A14) we obtain $\tau_x = \tau_y$.

These two conditions can also be deduced by the comparison of scaling relations that are derived from different coefficients, e.g., $\Theta_{I,1}, \Theta_{II,2}$ and $\Theta_{III,1}$. The same procedure can be carried out for the $y$-component of the SSA leading to the same outcome due to the symmetry of both horizontal components of the SSA. Applying our findings it follows $\Phi_x = \Phi$ in Eq. (A2) and the dimensionless ITE is identical to Eq. 27 with the same coefficient $\Omega$.

We thus found that in order to fulfill the required scaling similarity in the considered two-dimensional SSA-case there can only exist one horizontal length scale and one time scale (as opposed to one in each horizontal direction, as assumed initially). All the scaling relations derived for the flow-line SSA case (Eqs. 11 - 13) hold here.

## Appendix B: Experimental design of numerical simulations

### B1 Flow-line simulations

For the two-dimensional simulations, we use the symmetric flow-line geometry and the sequence of experiments described in Feldmann and Levermann (2015a): An ice sheet in equilibrium (grey profile in Fig.-2) is perturbed in its RHS basin, forcing the grounding line to retreat onto the basin's inward down-sloping bed section (Fig.-2, red profile). After cessation of the perturbation the grounding line continues to retreat indicating that a MISI has been triggered. The resulting far-inland spreading dynamic ice-sheet thinning eventually initiates a second MISI in the connected LHS basin (see Feldmann and Levermann, 2015a, for a detailed examination of the mechanism which is visualized in their Fig. 4a). This second MISI is induced only through internal ice dynamics without any direct forcing and hence we expect that the speed of the instability is a suitable measure to reflect the ice-sheet inherent response time.

For three parameter sets the simulations deviate from the above described scenario. In two simulations ($2D_{\alpha=1,\beta=\frac{1}{2},\delta=1}$ and $2D_{\alpha=\frac{1}{2},\beta=1,\gamma=1}$) the ice sheet does not find a steady state with a grounding-line location on the ocean side of the coastal sill but collapses after several thousand years and equilibrates on the central bed portion. In simulation $2D_{\alpha=1,\beta=\frac{1}{2},\gamma=1}$ only the first MISI is triggered but not the second (referred to as "stable" scenario S in Feldmann and Levermann, 2015a). Though the unstable retreat in these three simulations does not take place as unperturbed as in the scenario described further above we nevertheless use the speed of retreat to estimate ice-sheet response time also for these scaled setups.

### B2 Three-dimensional simulations

For the three-dimensional experiments we extend our flow-line geometry by introducing a second horizontal dimension ($y$) to obtain channel-like ice-sheet flow in three dimensions





with similar geometry as in Gudmundsson et al. (2012) and
Asay-Davis et al. (2015). The bed topography $b(x,y)$ is a
superposition of two components: The bed component in $x$-
direction, $b_x(x)$, is as described in Feldmann and Levermann
(2015a) but lowered uniformly by -300 m (Fig. 3). The com-
ponent in $y$-direction, $b_y(y)$, is taken from Gudmundsson
et al. (2012). The superposition of both, $b(x,y) = b_x + b_y$,
yields a bed trough which is symmetric in both $x$- and $y$-
direction (Fig. 4). While the main ice-sheet flow is still in
$x$-direction (from the interior through the bed trough towards
the ocean) there is also a flow component in $y$-direction, i.e.,
from the channel's lateral ridges down into the trough. Re-
sulting convergent flow and associated horizontal shearing
enable the emergence of buttressing, and hence ice dynamics
in this setup differ substanitally from the flow-line case. In
particular the buttressing effect stabilizes the grounding line
further downstream than would be expected in a flow-line
configuration (compare Figs. 2 to 3 where the steady-state
grounding lines are approximately at the same position but
the local bed elevation differs by several 100 m).

Spinning up the model we obtain a symmetric ice sheet in
equilibrium with a stable bay-shaped grounding line. Along
the centerline of the setup ($y = 0$) the grounding line is lo-
cated downstream of the coastal sill, similar to the flow-line
case (Figs. 4 and 3 in grey). Two symmetric ice shelves have
formed which are fringed and fed by ice from the inland and
lateral direction. The steady-state ice sheet is then perturbed
by removing all floating ice instantaniously after which a
continuous elemition of all ice that crosses the grounding
line is applied. This scaling-independent perturbation initi-
ates grounding-line retreat onto the inland-downsloping bed
and the synchronously unfolding MISIs provide a measure
for the ice-sheet response time.

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




**Table 1.** Parameter values as prescribed in the unscaled reference simulations for the flow-line setup (2D) and the three-dimensional channel setup (3D), respectively. For the scaling experiments the bed geometry ($b_x$ and $b_y$) and the parameters $a$, $A$ and $C$ are multiplied with the scaling ratios from to Table **??**. The terms "BC-300" and "BC0" refer to the bed geometries described in Feldmann and Levermann (2015a) and $b_{y,G}$ refers to $y$-component of the bed topography used in Gudmundsson et al. (2012).

| Parameter | 2D$_{ref}$ | 3D$_{ref}$ | Unit | Physical meaning |
|---|---|---|---|---|
| $a$ | 0.6 | 0.5 | $\mathrm{m\,yr^{-1}}$ | Surface mass balance |
| $b_x$ | "BC-300" | "BC0" - 300 m | | $x$-component of bed topography |
| $b_y$ | - | $b_{y,G}$ | | $y$-component of bed topography |
| $A$ | $10^{-25}$ | | $\mathrm{Pa^{-3}s^{-1}}$ | Ice softness |
| $C$ | $10^7$ | | $\mathrm{Pa\,m^{-1/3}s^{1/3}}$ | Basal friction parameter |
| $g$ | 9.81 | | $\mathrm{m\,s^{-2}}$ | Gravitational acceleration |
| $m$ | 1/3 | | | Basal friction exponent |
| $n$ | 3 | | | Exponent in Glen's law |
| $\rho_i$ | 900 | | $\mathrm{kg\,m^{-3}}$ | Ice density |
| $\rho_w$ | 1000 | | $\mathrm{kg\,m^{-3}}$ | Sea-water density |

**Table 2.** Scaling ratios as used for our numerical simulations. Prescribed scaling ratios are highlighted in blue, the other result from Eqs. (11)-(17), (29) and (30). Each row corresponds to a scaling experiment, that is carried out in flow line ("2D") and in a three-dimensional channel setup ("3D"). The parameters values prescribed in the simulations are obtained by multiplying $b_x$, $b_y$, $C$, $a$ and $A$ (see Table 1) with the given ratios $\alpha$, $\beta$, $\gamma$, $\delta$ and $\zeta$. The analytic values for $\dot\chi$ are used to fit the sections of linear grounding-line retreat in Figs. 5 and 6.

| Simulation name | $\alpha$ | $\beta$ | $\gamma$ | $\delta$ | $\zeta$ | $\tau$ | $\dot\chi$ |
|---|---|---|---|---|---|---|---|
| 2D/3D$_{REF}$ | 1 | 1 | 1 | 1 | 1 | 1 | 1 |
| 2D/3D$_{\alpha=1,\beta=\frac12,\gamma=1}$ | 1 | $\frac12$ | 1 | $\frac{1}{128}$ | $\frac18$ | 64 | $\frac{1}{64}$ |
| 2D/3D$_{\alpha=\frac12,\beta=\frac12,\gamma=1}$ | $\frac12$ | $\frac12$ | 1 | $\frac18$ | 2 | 4 | $\frac18$ |
| 2D/3D$_{\alpha=\frac12,\beta=1,\gamma=1}$ | $\frac12$ | 1 | 1 | 16 | 16 | $\frac{1}{16}$ | 8 |
| 2D/3D$_{\alpha=1,\beta=\frac12,\delta=1}$ | 1 | $\frac12$ | $\left(\frac12\right)^{7/3}$ | 1 | 16 | $\frac12$ | 2 |
| 2D/3D$_{\alpha=\frac12,\beta=\frac12,\delta=1}$ | $\frac12$ | $\frac12$ | $\frac12$ | 1 | 16 | $\frac12$ | 1 |
| 2D/3D$_{\alpha=\frac12,\beta=1,\delta=1}$ | $\frac12$ | 1 | $\left(\frac12\right)^{-4/3}$ | 1 | 1 | 1 | $\frac12$ |

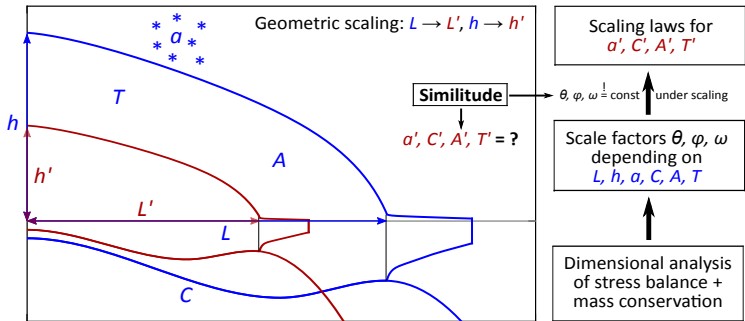

**Figure 1.** Schematic of the similitude-analysis method carried out in this study. A reference system (blue ice sheet and bed topography) with geometric scales $h$ and $L$, time scale $T$ and physical parameters ice softness $A$, basal friction coefficient $C$ and surface mass balance $a$ is scaled in horizontal and vertical direction (red contours, primed system). The goal is to derive the scaled parameters of the primed system under which dynamic similarity between both ice sheets holds. A dimensional analysis of the governing equations yields dimensionless scale factors which have to remain constant under scaling to attain similitude. The resulting scaling laws determine the scaled (primed) parameters.




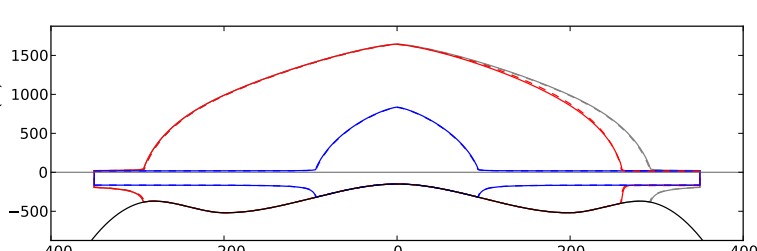

**Figure 2.** Ice sheet profiles at three different stages of the flow-line simulations $2D_{\alpha=\frac{1}{2},\beta=\frac{1}{2},\gamma=1}$ (continuous) and $2D_{ref}$ (dashed). Output of the reference simulation is scaled by factor 0.5 in both horizontal and vertical direction to allow for comparison of shapes between the two simulations.

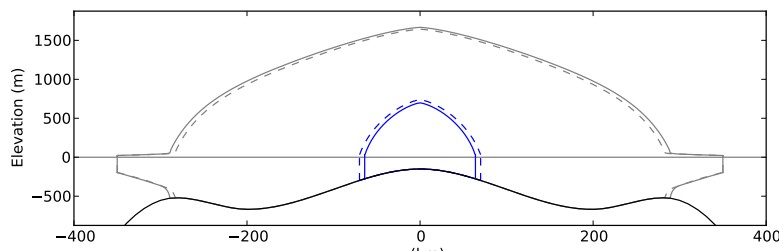

**Figure 3.** Steady-state ice-sheet profiles for cross section along the centerline ($y = 0$) of the three-dimensional channel setup for simulations $3D_{\alpha=\frac{1}{2},\beta=\frac{1}{2},\gamma=1}$ (continuous) and $3D_{ref}$ (dashed). Output of the reference simulation is scaled by 0.5 in both horizontal and vertical direction to allow for comparison of shapes between the two simulations.

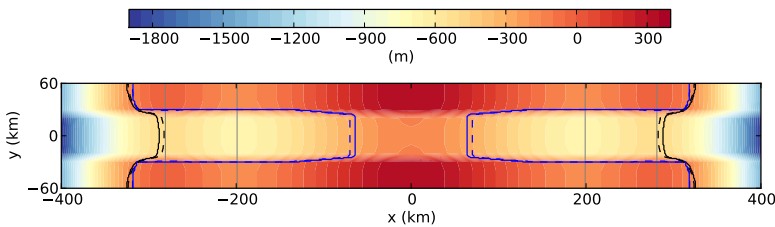

**Figure 4.** Bed topography of the three-dimensional channel setup, here shown in the scaled version with $\alpha = \beta = 0.5$ (see Fig. 3 for a cross section along $y = 0$). Steady-state grounding-line positions for simulations $3D_{\alpha=\frac{1}{2},\beta=\frac{1}{2},\gamma=1}$ (continuous) and $3D_{ref}$ (dashed). Grey lines mark the position of the coastal sill and the bed depression, respectively. Output of the reference simulation is scaled by 0.5 in horizontal direction to allow for comparison of shapes between the two simulations.





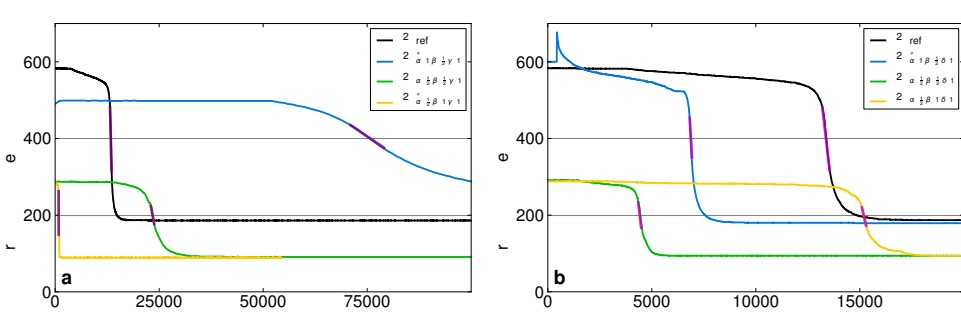

**Figure 5.** Time series of grounding-line position for the reference and three geometrically scaled flow-line experiments for which **(a)** basal friction and **(b)** surface mass balance is held constant, respectively. Grey horizontal lines indicate location of the minimum of the bed depression for both the scaled und unscaled case around which the grounding line retreats unstable and retreat rates are approximately constant. In this range the slope of the curve of the unscaled simulation is fitted to obtain a reference retreat rate of 0.47 km/yr (purple slope fitted to black curve) which is used to predict the slopes, i.e., retreat rates, for the scaled experiments (other curves overlayn by purple lines with predicted slopes) according to Table 1.

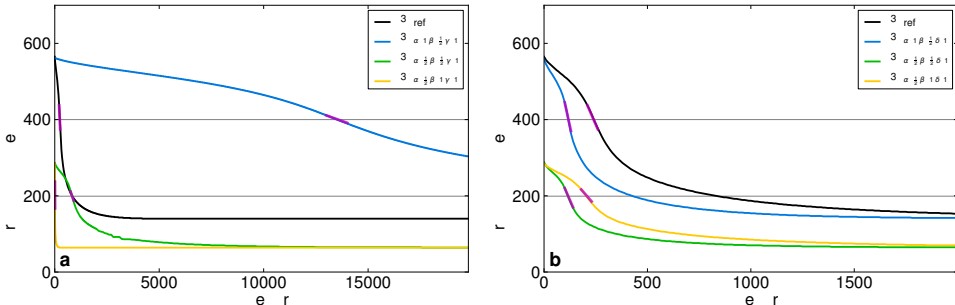

**Figure 6.** Time series of centerline grounding-line position (along $y = 0$) for the reference and three geometrically scaled 3D channel experiments for which **(a)** basal friction and **(b)** surface mass balance is held constant, respectively. The Fitting method is the same as described in Fig. 5



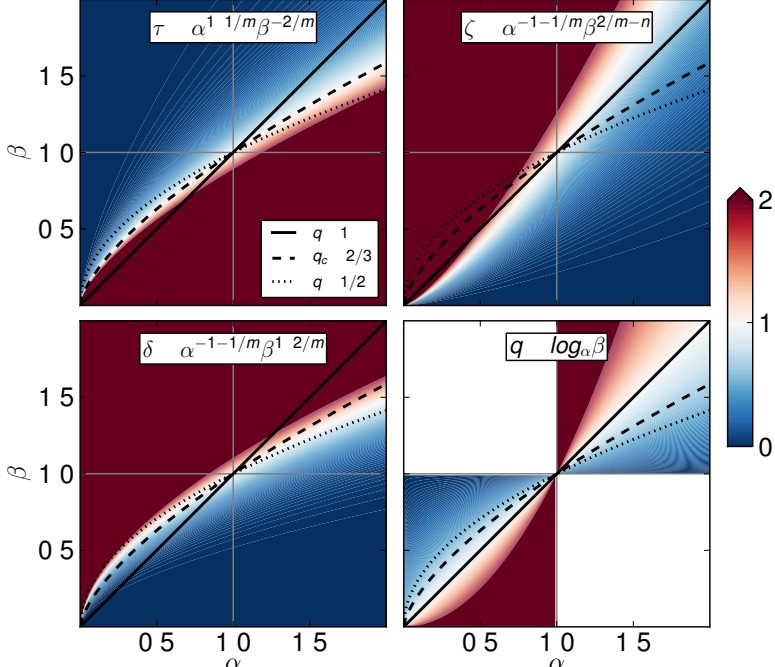

**Figure 7.** Scaling of **(a)** response time (Eq. 11), **(b)** ice softness (Eq. 16) and **(c)** surface mass balance (Eq. 15) in the $\alpha$-$\beta$ phase space for $\gamma = 1$ and $m = 1/3$. Panel **(d)** shows value of the exponent $q$ if the two horizontal scales are linked according to Eq. 32. Dotted line represents scaling of an ice sheet with Vialov proportions. Dashed line denotes critical threshold $\tau = 1$.





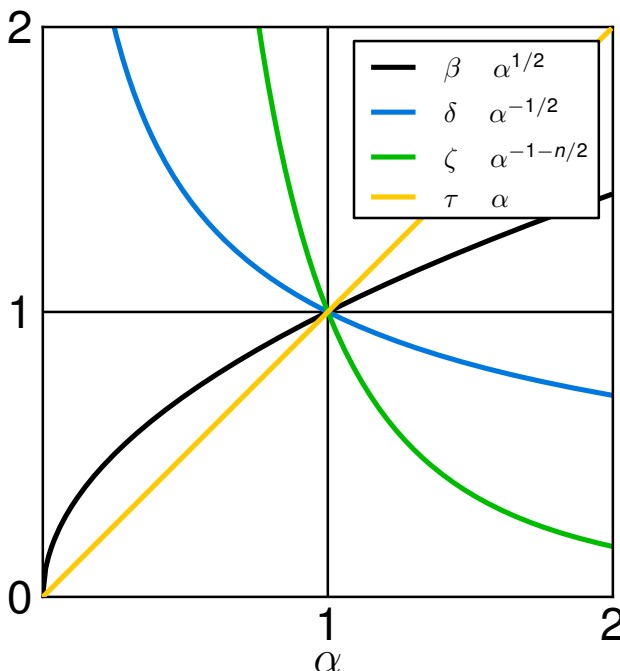

**Figure 8.** Scaling of response time $\tau$, surface mass balance $\delta$ and ice softness $\zeta$ under the assumption of Vialov-type geometric scaling ($\beta = \alpha^{1/2}$) and constant basal friction ($\gamma = 1$). The resulting scaling conditions are independent of $m$ and given in the legend ($n=3$).





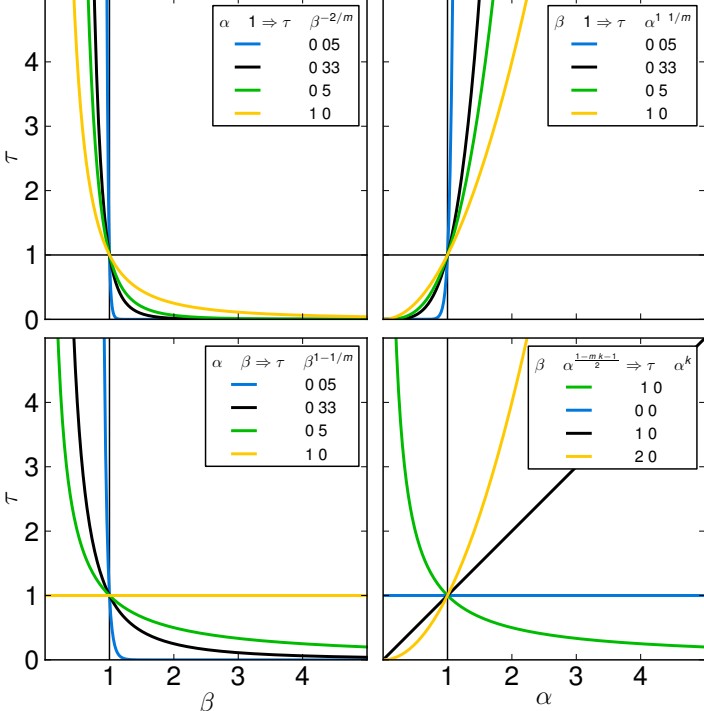

**Figure 9.** Response-time scaling for hypersurfaces through the $\alpha$-$\beta$-$m$ phase space according to Eq. 31 for **(a)** $\alpha = 1$, **(b)** $\beta = 1$, **(c)** $\alpha = \beta$ and **(d)** the constraint that the response time scales independently of $m$. In each panel the legend gives the scaling law for $\tau$ that results from the applied constraint.