# Peer review of "Similitude of ice-sheet dynamics against scaling of geometry and physical parameters"

_The Cryosphere, 2015_

## Referee Comment (RC1) · Anonymous Referee #1 · 12 Feb 2016

**The Cryosphere** - **TC2015-226** *"Similitude of ice-sheet dynamics against scaling of geometry and physical parameters"* by Feldmann and Levermann.

This paper presents a similitude analysis of the Shallow Shelf Approximation (SSA) pronostic equations. Such similitude analysis which seems commonly employed in other fields or research might have been ignored by glaciologist. This contribution is therefore interesting to see the potential of such method. In this paper, the method is validated against 2D and 3D numerical simulations. Greater impacts of the paper should certainly been expected by directly applying the method to real outlets glaciers of Antarctica or Greenland, but is certainly beyond the scope of this first paper and would certainly require further developments. This is overall a well written paper, even if it contains quite a lot of equations (which I was not able to verify all) and I would recommend its publication in TC. I have few remarks that are listed below.

**Remarks**

Abstract: the abstract is too long and should be shorten. There are repetitions from the abstract and introduction that could be avoided.

page 1, line 30: I haven't done this bibliography, but people working on flubber experiment as an analogue of ice must have had these questioning about the similitude of their experiment and a real glacier. By the way, similitude of analogue experiments is an other domain of application for the method that should be mentioned.

page 1, line 40 and below: I guess there are much more references than the one cited so I would suggest to use "e.g." in front of the references.

page 1, line 63: I don't get the point. Which has been shown to what?

page 2, line 14: I don't understand what you mean by "which will be put to test in the forthcomingMISMIP+ intercomparison project"?

below Eq. (1): not all the notations introduced in this equation are explained (e.g. $A$).

Equation (4) is neglecting basal mass balance (basal melting). It should be mentioned.

page 3, line 20: the use of compression is confusing as compression could refer to the state of stress. Elongation/Shortening?

page 3, lines 49-51: I am not sure to clearly understand the two limits. Especially the case $\Phi >> \theta$ since the case of a frozen bed cannot be modeled assuming the SSA. Also, to which equations do you refer when you said "in which non of the stress balance terms are neglected"? In the SSA, this is already not true as it neglects stress regarding to the Stokes equations. This should be clarified.

page 4, line 30: it is not the length of the entire ice-sheet, but only the grounded part (upstream the GL).

page 4, line 29: integration of (Eq. 4) over -> integration of Eq. (4) over (and at other places in the manuscript)

page 4, lines 31-35: I am not sure to follow what is really demonstrated here and not sure to see where is the consistency with the BLT of Schoof. Indeed, the equations derived by the BLT are based on the SSA ones, so that intuitively I would said that the similitude derived for the SSA also apply for the BLT? You should present it the other way, and derive directly the scaling relation (24)?

page 5, line 45: Is it really constant, which refer to time, whereas here one wants to said that it is the same value of the friction in the two experiments. "Same" or "identical" is may be better than "constant"? It should be modified accordingly all along the manuscript.

page 5, line 48: reference to Table 2 is broken

page 5, line 53: it should be mentioned here that the bedrock also varies in the transverse direction.

page 6, line 26: Vialov profiles are derived assuming the Shallow Ice Approximation (SIA) whereas here the SSA is used. Only in the conclusion it is mentioned that in a previous paper you have shown that SSA was conducting to similar profiles as Vialov ones. It should be mentioned here.

page 6, line 71: atmosphere: Rising -> atmosphere: rising (and at other places in the manuscript)

page 6, line 82: space (Fig. 7 accounts for only one value of m. -> space (Fig. 7 accounts for only one value of m).

page 7, line 22: To what refers "respectively"?

page 7, line 33: I don't understand what you mean here as you have already started from the SSA equations and not the full Stokes system of equations. There is a missing citation.

page 7, line 40: again, $\epsilon$ is used to derive the SSA from the Stokes equations so it has somehow been used already in the equations you are using here. This part is a bit confusing and would require some clarifications.

page 7, line 50: As already mentioned, I would said, but may be I misunderstood something, that this is normal as these BLT equations are derived from the SSA ones...

page 7, line 103: law still then still depends on -> law then still depends on

page 7, line 106: of m (9): Vertical -> of m (9): vertical

page 8, line 18: reasonably - and this should be said before.

page 8, lines 37-38: consider rewording and also avoid the repetition for the value of $n$.

page 10, line 33: to Eq. 27 with -> to Eq. (27) with

B1: define what is RHS and LHS

page 11, line 28: instantaneously, elimination

Figures 5 and 6: legend and axis label are not correct. Why not applying a scaling along x and t? How do you choose the part of the curve where is made the retreat rate comparison? In the legend: overlayn -> overlaid

legend Fig. 7: to Eq. 32. -> to Eq. (32).

legend Fig. 9: to Eq. 31 for -> to Eq. (31) for

---

## Referee Comment (RC2) · Anonymous Referee #2 · 27 Mar 2016

**Review of a manuscript "Similitude of ice-sheet dynamics against scaling of geometry and physical parameters " by J. Feldmann and A. Levermann.**

The manuscript presents similarity solutions for the isothermal Shallow Shelf Approximation (SSA) equations. Though, to my knowledge, such solutions for the SSA have not been derived before, the manuscript has a number of conceptual inconsistencies and cannot be published in its present form.

**Major concerns**

The first major concern is an assumption that ice is isothermal and the independence of the ice softness parameter of other parameters, e.g. ice thickness or surface mass-balance. Thicker ice is usually softer than thinner ice, hence more deformable. Physically, $A^{-1/n}$ decreases with increasing ice thickness. The constant $\theta$ (eqn. 9) implies the opposite. Though, mathematically there is nothing wrong with this assumption, the derived similarity solutions are not suitable for glaciological applications. One possibility to resolve this inconsistency could be to consider temperature itself (or depth-averaged or depth-integrated temperature) instead of the ice softness parameter $A$. It still can be spatially uniform in the horizontal direction, but vary with an ice-stream of ice-shelf characteristic thickness.

The second major concern is the chosen dependence of the surface mass-balance ratio $\delta$ on the friction-coefficient ratio $\gamma$ (eqn. 15). Physically, the surface mass-balance depends on a climate, and has no connection to ice-stream properties like basal friction. Though, there is a connection between the basal friction coefficient and the ice stiffness parameter (eqn. 16), it is very weak, as frictional heating affects ice temperature, hence its stiffness, only very small part of the ice column, close to its bottom.

There is no relevance of the similarity solutions derived in this study to the Shoof's (2007) boundary layer theory. The system of equations considered in both studies is the same, so it is not

surprising that the flux formulation is identical (in a non-dimensional form).

Throughout the text, the described ice flow is referred to as "ice-sheet" flow. This is misleading, because the SSA equations are valid only in ice-stream and ice-shelf settings, and are inapplicable to the rest of an ice sheet. Equally, the use of the Vialov profile (even as a motivation) is incorrect. This profile is derived based on the Shallow Ice Approximation (SIA).

The mass-conservation equation (4), for some reasons called in this studythe "ice thickness equation (ITE)", omits the basal mass-balance. This may be appropriate for ice streams, however, on ice shelves with strong basal melting neglecting this term is incorrect.

The abstract implies that the similarity analysis has never been applied in glaciology. This is not true; Halfar (19833) and Buler et al. (2005) describe similarity solutions for various configurations of the SIA.

**Minor concerns**

P. 3 Line 20 $\alpha, \beta < 1$ suggests that $\alpha$ and $\beta$ can be negative.

P.3 Lines 41-51. Sentences starting with "In Eq. (9)..." do not make sense. A statement "$\psi \ll \theta$, holding for ice frozen to bedrock" is incorrect. The SSA is inapplicable in circumstances where ice is frozen to bedrock.

P. 3 eqn (14) and P.4 eqn (17), though mathematically are correct, physically not so. The ice softness and surface mass balance are unlikely scale identically. The basal friction is independent (to a leading order) of the surface mass-balance.

P.5 Lines 50-55. The numerical simulations are one-dimensional (a flow-line setup) and two-dimensional (channel-flow setup). The SSA are vertically integrated equations and do not have vertical dimension.

In many places references are missing (e.g. p.7 line 34).

In summary, the presented similarity analysis of the SSA equations has little to do with ice-stream and ice-shelf flow. The derived set of the dimensionless parameters is perfectly fine for an abstract

set of equations of the SSA form. However, shortcomings of the study described above most likely lead to erroneous conclusions of the similarity behaviour of the ice streams, ice shelves and grounding lines.

**Rerences**

Bueler, E., Lingle, C. S., Kallen-Brown, J. A., Covey, D. N., Bowman, L. N. (2005), Exact solutions and verification of numerical models for isothermal ice sheets, J. Glac., 51(173), 291-306, doi:10.3189/172756505781829449.

Halfar P. (1983), On the Dynamics of the Ice Sheets 2 J. Geophys. Res., 88(C10), 6043-60, doi:10.1029/2011JF002246.

---

## Author Comment (AC1) · 24 May 2016

**Detailed response to the editor on manuscript tc-2015-226**

"Similitude of ice-sheet dynamics against scaling of geometry and physical parameters"

by J. Feldmann and A. Levermann

Dear Dr Pattyn,

We would like to thank you for handling the review process and the reviewers for their detailed look at our manuscript. We are happy with the first reviewer's assessment that our analysis is very fundamental in nature, that has been carried out for other field theories and will become very helpful in future glaciological applications. We would like to thank the reviewer for this very constructive and very positive response. Two figures have been added to the manuscript (Figs. C1 and 2), following the suggestion of reviewer #1.

Unfortunately we detected a misunderstanding by reviewer #2 of our study. We have elaborated on the purpose of our analysis in the detailed response and hope that the editor agrees with us that the analysis is not as useless as reviewer #2 is convinced it is. It would be great if our explanations below would convince reviewer #2 that we provide a useful theoretical analysis without being able to claim comprehensiveness.

Please find below the *reviewers' comments in italics* and our detailed response in blue. We have further attached a revised manuscript that highlights the changes in the submission, as well as a clean revised version.

Best wishes,
J. Feldmann and A. Levermann

**Reviewer #1:**

***The Cryosphere - TC2015-226*** *"Similitude of ice-sheet dynamics against scaling of geometry and physical parameters" by Feldmann and Levermann.*

*This paper presents a similitude analysis of the Shallow Shelf Approximation (SSA) prognostic equations. Such similitude analysis which seems commonly employed in other fields or research might have been ignored by glaciologist. This contribution is therefore interesting to see the potential of such method. In this paper, the method is validated against 2D and 3D numerical simulations. Greater impacts of the paper should certainly been expected by directly applying the method to real outlets glaciers of Antarctica or Greenland, but is certainly beyond the scope of this*

*first paper and would certainly require further developments. This is overall a well written paper, even if it contains quite a lot of equations (which I was not able to verify all) and I would recommend its publication in TC. I have few remarks that are listed below.*

Response: We would like to thank the reviewer for the readiness to review our manuscript. The reviewer comments were very constructive and helpful in improving our manuscript.

*Abstract: the abstract is too long and should be shorten. There are repetitions from the abstract and introduction that could be avoided.*

Response: We shortened the abstract to avoid repetitions in abstract and introduction.

*page 1, line 30: I haven't done this bibliography, but people working on flubber experiment as an analogue of ice must have had these questioning about the similitude of their experiment and a real glacier.   By the way, similitude of analogue experiments is an other domain of application for the method that should be mentioned.*

Response: This was a very useful hint. We now mention the application of similitude analysis in laboratory glacier experiments in the abstract (page 1, lines 25-26) and in the introduction (page 1, lines 49-54) of our manuscript.

*page 1, line 40 and below: I guess there are much more references than the one cited so I would suggest to use "e.g." in front of the references.*

Response: We agree and added "e.g." here and in front of the next references.

*page 1, line 63: I don't get the point. Which has been shown to what?*

Response: We are sorry for the lack of clarity and have tried to make this clearer (page 2, lines 11-14).

*page 2, line 14: I don't understand what you mean by "which will be put to test in the forthcoming MISMIP+ intercomparison project"?*

Response: This phrase is indeed out-of-context and thus we deleted it.

*below Eq. (1): not all the notations introduced in this equation are explained (e.g. A).*

Response: Thanks for the hint. We indeed missed to introduce ice density $\rho$ and gravitational acceleration $g$ and added them below Eq. (1) (page 2, lines 92-93). Ice softness $A$ was already introduced in the line above Eq. (1).

*Equation (4) is neglecting basal mass balance (basal melting). It should be mentioned.*

Response: That's true. We added a sentence below Eq. (4) (page 3, lines 20-23).

*page 3, line 20: the use of compression is confusing as compression could refer to the state of stress. Elongation/Shortening?*

Response: We are glad for the reviewer's suggestion and replaced the terms here and elsewhere in

the manuscript.

*page 3, lines 49-51: I am not sure to clearly understand the two limits. Especially the case*
*Φ >> θ since the case of a frozen bed cannot be modeled assuming the SSA. Also, to which*
*equations do you refer when you said "in which non of the stress balance terms are neglected"?*
*In the SSA, this is already not true as it neglects stress regarding to the Stokes equations.*
*This should be clarified.*

Response: These lines were indeed misleading (especially the statement about the $Φ >> θ$ limit is incorrect, as noted by the reviewer). We re-formulated the paragraph accordingly (page 3, lines 54-59).

*page 4, line 30: it is not the length of the entire ice-sheet, but only the grounded part*
*(upstream the GL).*

Response: We thank the reviewer for the hint and corrected the line.

*page 4, line 29: integration of (Eq. 4) over -> integration of Eq. (4) over (and at other*
*places in the manuscript)*

Response: Done. At other places "(Eq. x)" is intended to avoid double brackets, i.e. (Eq. (x)), see, e.g. page 4, line 81. If applicable, we will be glad to change the notation according to the TC conventions in the final typesetting phase.

*page 4, lines 31-35: I am not sure to follow what is really demonstrated here and not sure to*
*see where is the consistency with the BLT of Schoof. Indeed, the equations derived by the*
*BLT are based on the SSA ones, so that intuitively I would said that the similitude derived for*
*the SSA also apply for the BLT? You should present it the other way, and derive directly the*
*scaling relation (24)?*

Response: This remark was indeed second by the second reviewer and we really appreciate that the reviewers demand that we reconsider this section. Please consider the following: The Shallow Shelf Approximation is a non-linear differential equation and the solution by Christian Schoof required a few additional assumptions, for example he omitted the membrane stresses in the stress balance and reintroduced them in the boundary conditions. He also does not consider a time dependence which we reintroduce in the commonly used way via the mass continuity equation. Given these differences we think it is not completely trivial that the SSA-scaling that we derive survives all the way to the final solution given by Schoof (2007b). Obviously it has and that makes it almost look trivial again. In any case we tend to think that it is at least a nice illustration of the scaling in one of the solutions of the SSA equation which is a simpler equation than the full SSA. We would be willing to omit this section if demanded by the editor and reviewers, but we would prefer to keep it in, if possible.

*page 5, line 45: Is it really constant, which refer to time, whereas here one wants to said that*
*it is the same value of the friction in the two experiments. "Same" or "identical" is may be*

*better than "constant"? It should be modified accordingly all along the manuscript.*

Response: We are glad for the valuable hint and changed the terms here and in the rest of the manuscript.

*page 5, line 48: reference to Table 2 is broken*

Response: Done.

*page 5, line 53: it should be mentioned here that the bedrock also varies in the transverse direction.*

Response: Done.

*page 6, line 26: Vialov profiles are derived assuming the Shallow Ice Approximation (SIA) whereas here the SSA is used. Only in the conclusion it is mentioned that in a previous paper you have shown that SSA was conducting to similar profiles as Vialov ones. It should be mentioned here.*

Response: Done.

*page 6, line 71: atmosphere: Rising -> atmosphere: rising (and at other places in the manuscript)*

Response: Done.

*page 6, line 82: space (Fig. 7 accounts for only one value of m. -> space (Fig. 7 accounts for only one value of m).*

Response: Done.

*page 7, line 22: To what refers "respectively"?*

Response: Corrected.

*page 7, line 33: I don't understand what you mean here as you have already started from the SSA equations and not the full Stokes system of equations. There is a missing citation.*
*page 7, line 40: again, is used to derive the SSA from the Stokes equations so it has somehow been used already in the equations you are using here. This part is a bit confusing and would require some clarifications*

Response: We thank the reviewer for the careful reading. We re-worded the paragraph (page 7, lines 60-71) to be more precise about what we want to say and in particular make clear that we consider the **SSA** stress balance here (previously we somewhat sloppily only wrote stress balance).

*page 7, line 50: As already mentioned, I would said, but may be I misunderstood something, that this is normal as these BLT equations are derived from the SSA ones...*

Response: Please see our response to the above comment related to page 4 lines 31-35.

*page 7, line 103: law still then still depends on -> law then still depends on*
Response: Done.

*page 7, line 106: of m (9): Vertical -> of m (9): vertical*
Response: Done.

*page 8, line 18: reasonably - and this should be said before.*
Response: Done. Also see above.

*page 8, lines 37-38: consider rewording and also avoid the repetition for the value of n.*
Response: Done.

*page 10, line 33: to Eq. 27 with -> to Eq. (27) with*
Response: Done.

*B1: define what is RHS and LHS*
Response: Done.

*page 11, line 28: instantaneously, elimination*
Response: Done.

*Figures 5 and 6: legend and axis label are not correct.*
Response: These must have got broken during the TCD publishing phase since the original submitted file had correct labels and legends as is the case for the revised manuscript.

*Why not applying a scaling along x and t? How do you choose the part of the curve where is made the retreat rate comparison?*
Response: We thank the reviewer for the useful hint to scale the time series along both axis. In the scaled plots the curves of grounding-line position during the ice-sheet instability collapse approximately into a single curve, indicating similitude between the experiments as expected from theory (Figs. C1 and C2). At the same time it is most important to us to visualize the variety of retreat rates (slopes) during unstable grounding-line retreat simulated in our scaling experiments (Figs. 5 and 6). We thus are in favor of keeping the original unscaled plots in the results section and included the scaled plots into the Appendix (Appendix C).
The range, i.e. the section of the curve, for the retreat-rate comparison (visualized in Figs. 5 and 6) was chosen by hand originally. We now use an objective criterion to define that range (x-range +-50 km around the minimum of the bed depression) within which we fit the slopes of approximately constant grounding-line retreat. We updated Figs. 5 and 6 and their captions accordingly.

*In the legend: overlayn -> overlaid*
Response: Done.

*legend Fig. 7: to Eq. 32. -> to Eq. (32).*

Response: Done.

*legend Fig. 9: to Eq. 31 for -> to Eq. (31) for*

Response: Done.

**Reviewer #2:**

**Review of a manuscript "Similitude of ice-sheet dynamics against scaling of geometry and physical parameters " by J. Feldmann and A. Levermann.**

*The manuscript presents similarity solutions for the isothermal Shallow Shelf Approximation (SSA) equations. Though, to my knowledge, such solutions for the SSA have not been derived before, the manuscript has a number of conceptual inconsistencies and cannot be published in its present form.*

Response: This criticism is in stark contrast to the assessment of the first reviewer and as we will show below the criticism is not substantiated by the reviewer. The requests made by the reviewer are by large extensions of our work. We, however, agree with the first reviewer that our manuscript is indeed already quite long and has a proper scope. We would like to emphasize that we do not attempt to give a comprehensive analysis of glaciers, but simply apply a very fundamental method to one of the two most commonly used approximations of the primitive equations of ice dynamics. We agree with the first reviewer that this analysis will turn out to be helpful in a number of future applications. Our analysis is based on a number of assumptions that are clearly and transparently emphasized in the manuscript. For example we assume an isothermal glacier. This is an obvious reduction of generality of our analysis, but it does not render it useless. In fact reduction in generality always occurs when an approximation is applied. As an example, the Shallow Ice Approximation is only rigorously valid when the ice is frozen to the ground which is not necessarily true in ice streams, but despite this constrain on the applicability of the approximation, it turned out to be very useful in glaciological theory. While we appreciate the comments made by the reviewer and have used them to improve our manuscript where this was possible, we would appreciate if we could keep the core of our analysis as it is.

***Major concerns***

*The first major concern is an assumption that ice is isothermal and the independence of the ice softness parameter of other parameters, e.g. ice thickness or surface mass-balance. Thicker ice is usually softer than thinner ice, hence more deformable. Physically, $A-1/n$ decreases with increasing ice thickness. The constant $\vartheta$ (eqn. 9) implies the opposite. Though, mathematically there is nothing wrong with this assumption, the derived similarity solutions are not suitable for glaciological applications. One possibility to resolve this inconsistency could be to consider temperature itself (or depth-averaged or depth-integrated temperature) instead of the ice softness parameter A. It still can be spatially uniform in the horizontal direction, but vary with an ice-stream of ice-shelf characteristic thickness.*

Response: We appreciate the reviewer's concern. Perhaps we have not made it clear enough that the theory that we derive is only valid under certain assumptions. The assumption of isothermal

ice was mentioned in the abstract and a number of times in the manuscript itself. There is a vast amount of glaciological theory published that assumes isothermal ice (see added references on page 2, lines 7-9). It is a strong assumption which is not always valid but it allows for insights into other aspects of ice dynamics that do not depend strongly on the spatial and temporal thermal structure of the ice. Obviously the temperature dependence of the softness is accounted for in these kind of theories, but the changes with space and time are neglected. We have now changed the introduction (page 2, lines 4-9) as well as the discussion (page 8, lines 90-103) to make it clearer that we assume isothermal ice and that our conclusions are only valid within this restriction. Future analysis not assuming isothermal ice but allowing for a spatial distribution of temperature within the ice might be interesting even though in that case it would have to be decided which is a generic spatial structure that can be assumed without restricting the results too strongly. In order to get results that fully integrate the thermal evolution of the ice sheet, numerical models are of course available.

At this stage, we would appreciate if the editor and reviewer would allow us to keep the analysis restricted to isothermal ice after we have now further emphasized that this is a restriction..

*The second major concern is the chosen dependence of the surface mass-balance ratio δ on the friction-coefficient ratio γ (eqn. 15). Physically, the surface mass-balance depends on a climate, and has no connection to ice-stream properties like basal friction. Though, there is a connection between the basal friction coefficient and the ice stiffness parameter (eqn. 16), it is very weak, as frictional heating affects ice temperature, hence its stiffness, only very small part of the ice column, close to its bottom.*

Response: This comment by the reviewer leads us to suspect that the reviewer has not fully understood the idea of scaling analysis in general and our analysis in particular. We do, of course, never assume that the surface mass balance ratio depends on the friction coefficient. We would have no grounds for that. The equation that the reviewer is referring to is a **result** as opposed to an assumption. Perhaps it helps to paraphrase the spirit of these kind of scaling analysis as follows: **If** two ice sheets are self-similar **then** these relations have to be fulfilled. That is to say: if for example you find a glacier with a certain ice thickness and a certain bed slope etc. and if this glacier has the same qualitative profile as another one which however has a different thickness and a different bed slope etc. then our analysis shows (for isothermal ice) that the basal friction and the surface mass balance need to have a specific relation otherwise this glacier cannot be in equilibrium with its environment and the SSA equation. We are sorry that our manuscript obviously was not clear enough for the reviewer to understand this point and are glad that the first reviewer understood the concept. We have tried to make this now clearer in the manuscript (page 7, lines 94-104).

*There is no relevance of the similarity solutions derived in this study to the Shoof's (2007) boundary layer theory. The system of equations considered in both studies is the same, so it is not surprising that the flux formulation is identical (in a non-dimensional form).*

Response: Please see our response to the same remark made by reviewer #1 in reference to page 4 lines 31-35.

*Throughout the text, the described ice flow is referred to as "ice-sheet" flow. This is misleading, because the SSA equations are valid only in ice-stream and ice-shelf settings, and are inapplicable to the rest of an ice sheet. Equally, the use of the Vialov profile (even as a motivation) is incorrect. This profile is derived based on the Shallow Ice Approximation (SIA).*

Response: The SSA has been used to describe the dynamics of ice sheets before, for example by Schoof (2007a), Goldberg et al. (2009), Gudmundsson et al. (2012), and whenever SSA-only modes are used in model intercomparisons. While it is true that the SSA has restrictions, for example that it can only be used for the depth-averaged horizontal flow, it is not true that it is a completely invalid representation of ice sheet flow. We however appreciate the reviewers concern and have changed the phrase "ice sheet flow" to "ice flow". I think that it is clear to the reader that we only analyze SSA dynamics. We have also made some changes in order to highlight that the Vialov profile is derived from the SIA equation (page 6, lines 47-52). In the manuscript we use it for comparison with an analytic result. We believe that this is enlightening for the reader and that omitting this comparison would be a shame. Since we do nothing unscientific or intransparent here, we would appreciate if we could keep it in.

*The mass-conservation equation (4), for some reasons called in this study the "ice thickness equation (ITE)", omits the basal mass-balance. This may be appropriate for ice streams, however, on ice shelves with strong basal melting neglecting this term is incorrect.*

Response: We are grateful for the reviewer's scrutiny, but again this comment is not relevant to our analysis. We never said that we study ice shelves in the presence of basal melt as there are a lot of things we do not study. Equivalent equations to those we derive here are present in every textbook of hydrodynamics for about 100 years (I personally own a copy of the book by Sir Horace Lamb from the 19[th] century which describes the similarity analysis for the Navier-Stokes equation). To our knowledge these equations are, however, not present in the glaciological literature for the SSA dynamics. Thus, while we do not claim to be comprehensive in our analysis in the sense that we have a theory for all ice dynamics, we believe that these equations make some scientific contribution in their present form and would be grateful if we could publish them without including basal melt. In order to avoid misunderstanding, we now make clear in the text where we define the ice thickness equation (Eq. 4) that throughout the study we focus on the grounded part of the ice sheet and that we neglect the basal mass balance (page 3, lines 21-24 and lines 57-58; page 4, lines 47-48). The term "ice thickness equation" is used frequently in glaciological textbooks and we would like to keep it here.

*The abstract implies that the similarity analysis has never been applied in glaciology. This is not true; Halfar (19833) and Buler et al. (2005) describe similarity solutions for various configurations of the SIA.*

Response: We thank the reviewer for this helpful advice. We modified the abstract and now provide examples for previous use of the similarity concept in the field of glaciology in the introduction, including the mentioned similarity solutions for the SIA and laboratory experiments

in which glacier flow is simulated using a replacement material for ice (page 1, lines 48-58).

**Minor concerns**

*P. 3 Line 20 α, β < 1 suggests that α and β can be negative.*
Response: We thank the reviewer for the hint and added 0 as the lower bound.

*P.3 Lines 41-51. Sentences starting with "In Eq. (9). . . " do not make sense. A statement "ψ >> ϑ, holding for ice frozen to bedrock" is incorrect. The SSA is inapplicable in circumstances where ice is frozen to bedrock.*
Response: We are grateful to the reviewer for pointing to this statement that is indeed incorrect in the SSA context. We re-formulated the paragraph.

*P. 3 eqn (14) and P.4 eqn (17), though mathematically are correct, physically not so. The ice softness and surface mass balance are unlikely scale identically. The basal friction is independent (to a leading order) of the surface mass-balance.*
Response: Please see our answer to the reviewers second "major concern". This is not a claim we make, it is a result of the similarity assumption and thus one of the consequences for "similar ice geometries". Since ice softness and surface mass balance can vary independently it would indeed be invalid (to put it mildly) to assume that they scale with each other, but we do not do this. We hope this has become clearer in the text now.

*P.5 Lines 50-55. The numerical simulations are one-dimensional (a flow-line setup) and two-dimensional (channel-flow setup). The SSA are vertically integrated equations and do not have vertical dimension.*
Response: That is correct and so is our analysis. We do not see a problem here, but perhaps we are missing the reviewer's point here.

*In many places references are missing (e.g. p.7 line 34).*
Response: Done.

*In summary, the presented similarity analysis of the SSA equations has little to do with ice-stream and ice-shelf flow. The derived set of the dimensionless parameters is perfectly fine for an abstract set of equations of the SSA form. However, shortcomings of the study described above most likely lead to erroneous conclusions of the similarity behaviour of the ice streams, ice shelves and grounding lines.*
Response: We are happy that the reviewer assesses our similarity analysis to be mathematically "perfectly fine". In light of the fact that similar analysis for the Navier-Stokes equation have been used for decades to build airplanes and ships and have been proven tremendously helpful for the theoretical understanding of hydrodynamic flow, we are confident that our analysis is not completely useless. While we cannot hope to have similar impact on the glaciological community

as the hydrodynamics similarity analysis, we believe that this analysis is at least useful enough to be published.

[revised manuscript text omitted]

---

## Author Comment (AC2) · 24 May 2016

The comment was uploaded in the form of a supplement:
http://www.the-cryosphere-discuss.net/tc-2015-226/tc-2015-226-AC2-supplement.pdf

---

## Author Comment (AC3) · 25 May 2016

AC1 includes an old version of our response that was uploaded accidentally. Please see AC2 for the correct version of our response and use that for reference.
* * *

---

## Author Response (AR2)

**Detailed response to the editor on revised manuscript tc-2015-226**

"Similitude of ice dynamics against scaling of geometry and physical parameters"

by J. Feldmann and A. Levermann

Dear Prof. Pattyn,

We would like to thank you for the careful handling of the review process of our manuscript and are glad to hear that the manuscript might be suitable for publication after minor revisions.

We followed both of the recommendations made by the Editor: we moved Sec. 2.3, which shows consistency of our results with boundary-layer theory, to the Appendix. Also, we now clearly state in the discussion section that the approximation of SSA ice-sheet profiles with the SIA Vialov profile, which holds for our specific setup, should not be expected to apply in general due to the very distinct nature of the two approximations (page 8, lines 51-58). In addition, we slightly modified the manuscript title (replacing "ice-sheet dynamics" with "ice dynamics"), avoiding to suggest that our study addresses dynamics of a whole ice sheet with an SSA-based analysis.

Please find below the *reviewers' comments in italics* and our detailed response in blue. We have further attached a revised manuscript that highlights the changes in the submission, as well as a clean revised version.

Best wishes,
J. Feldmann and A. Levermann

**Reviewer #1:**

*I have read the new version of the paper as well as the reply of authors to both reviews. I am happy with this new version and have only noticed few minors typos.*

*- page 2, line 92: where vx and vy are the components in x and y direction of the horizontal velocity vector v, respectively, H...*
Response: Done

*- page 3, line 7: delete then the definition of the velocity vector here.*
Response: Done

*- page 3, line 12: behaviorS*
Response: Done

*- page 3, line 25: the fact that z is the vertical axis should be mentioned in the previous section.*
Response: Now mentioned in previous section (page 2, line 95 – page 3, line 1)

*- Figure 2 : give for which time are the different plot?*
Response: Done

**Reviewer #3:**

*This paper presents a dimensional analysis for the Shallow Shelf Approximation (SSA) equation, which is commonly used by glaciologists to model the ice dynamics of sliding dominant flow like those of tidewater glaciers and ice shelves. As a result, the authors obtain some relationships between geometrical, ice flow and mass balance parameters under the assumption that some dimensionless factors characterizing the flow are constant under scaling. Some of the relations are validated by numerical experiments.*

*This paper presents new theoretical results, which will certainly be of interest to ice flow modelers, but also a more general audience of glaciologists since the relationships derived in this paper can lead to new interpretations about the interaction between the main processes involved in ice sheets dynamics. In particular, the relation of the time response with respect to geometrical scaling factors, namely Eq. (31), is a key achievement, which justifies alone that this paper can be published in TC. As often, theoretical results are first obtained after simplifying the model set-up, and this gives some intuition on how strong the result is conditioned by the original assumptions, and whether these assumptions can be overcome or not. For this reason, and despite the fact that the applicability of isothermal SSA is very limited in real modelling cases, the assumptions made on flow (as isothermal) should not be an obstacle to the publication of this original work. In addition, I don't think that "thicker ice is usually softer than thinner ice" implies that "$A^{(1/n)}$ decreases with*

*the ice thickness". Softening is directly induced by an increase of strain-rate in response to thicker ice (Glen's law) independently of any change in the rate factor A. Finally, I don't think that the model assumes a priori any dependence between mass balance and friction. The dependence of the two comes a posteriori from the scaling analysis as expected.*

*With that said, I had a few (rather minor) concerns when reading the paper, which could potentially lead to some improvements in the final version.*

Response: We would like to thank the reviewer for the readiness to review our manuscript and are glad about the positive assessment and the constructive criticism.

*\* To me the scaling relationships (14)-(17) are the main achievement of the paper, and those must be emphasized. By contrast, the consistency with the boundary-layer theory is somehow expected (otherwise the computations would have been algebraically wrong) since this boundary-layer theory is also SSA-based. Thus, this is to me a weaker result, which is maybe over-stated in the paper.*

Response: We thank the reviewer for the advice to put more emphasis on the main findings of our study. We basically took this part out of the abstract and conclusions section to avoid overstating the consistency of our results with boundary-layer theory. As also suggested by the Editor, we moved original Sec. 2.3 on the boundary-layer theory to the Appendix (referred to on page 4, lines 20-22).

*\* If I understand correctly, Fig 5 and 6 are the (only) connection between theory and experimental (to validate (31)). By contrast, Fig. 7, 8 and 9 are simple plots of the relationships you derived in Section 2.2. This order of result's presentation is a bit disturbing since Fig. 7, 8 and 9 could have be drawn right after Section 2.2.*

Response: It is correct that Figs. 5 and 6 (and their alternative versions in the Appendix, Figs. D1 and 2) are the only plots which compare between theory (Sec. 2) and simulations (Sec. 3). Fig. 5 compares the time scaling according to Eq. (11) for the special case of identical friction (=Eq. (31)). Fig. 6 compares the time scaling according to Eq. (13) with assumed identical surface mass balance. From our point of view Figs. 7, 8 and 9 cannot be shown earlier as the reviewer suggests. The reason is that all three figures are based on further assumptions to Eq. (31) that are made in Sec. 4, i.e., specific values of $m$ (Figs. 7 and 9) and the introduced link between horizontal and vertical scale via exponent $q$=1/2 (Fig. 8). We understand the reviewers concerns, but at the moment see no other way of presentation and thus hope this is not too disturbing for the reader .

*\* Dimensional analysis are by nature demanding in term of numbers of variables to be introduced. However, I have the feeling that the paper can still be more efficient and more easily-readable by renaming variables in a more intuitive way so that the reader remembers more easily that, for instance, xi is related to sliding, delta is related to mass balance, ect.*

Response: We agree with the reviewer that it is demanding to find appropriate names for all of the numerous variables. While we think that it is appropriate to choose greek letters for the scaling

ratios and that these are intuitive for most of the ratios (alpha, beta, gamma, tau), the choice of variables delta and xi is indeed not fully satisfying. We thought hard about alternatives, but did not come up with better names. If that is ok we would like to keep it the way it is currently.

[revised manuscript text omitted]